# Allosteric activation unveils protein-mass modulation of ATP phosphoribosyltransferase product release
Benjamin J. Read [1], John B. O. Mitchell [2] & Rafael G. da Silva [1] ✉

Heavy-isotope substitution into enzymes slows down bond vibrations and may alter transition-state barrier crossing probability if this is coupled to fast protein motions. ATP phosphoribosyltransferase from *Acinetobacter baumannii* is a multi-protein complex where the regulatory protein HisZ allosterically enhances catalysis by the catalytic protein $HisG_S$. This is accompanied by a shift in rate-limiting step from chemistry to product release. Here we report that isotope-labelling of $HisG_S$ has no effect on the nonactivated reaction, which involves negative activation heat capacity, while HisZ-activated $HisG_S$ catalytic rate decreases in a strictly mass-dependent fashion across five different $HisG_S$ masses, at low temperatures. Surprisingly, the effect is not linked to the chemical step, but to fast motions governing product release in the activated enzyme. Disruption of a specific enzyme-product interaction abolishes the isotope effects. Results highlight how altered protein mass perturbs allosterically modulated thermal motions relevant to the catalytic cycle beyond the chemical step.

A sizable body of evidence point to contributions from protein dynamics to enzyme catalysis, spanning both slow, thermally equilibrated motions that redefine the enzyme conformational ensemble towards active populations[1–4], and non-statistical, femtosecond/picosecond-timescale vibrations coupled to transition-state barrier crossing[5–7]. In the context of allostery, where an enzyme's active site responds to perturbation (e.g., ligand binding) of a remote site in the protein, a role is often invoked for thermal motions governing ligand association/dissociation, product release, and cycling time among conformations[3,8], and, more recently, modulating the chemical step[2,9]. Yet it is challenging to separate such motions from those stemming from the inherent flexibility of proteins. In addition, experimental probes aiming to alter protein dynamics (e.g., replacement of specific residues) may also perturb the electrostatic potential surface of the system, making it difficult ascribe effects on rates solely to modulation of specific motions[10–12]. This difficulty arises from the significant rate enhancement achieved by substrate binding to an electrostatically preorganised active site, which minimises the reorganisation energy necessary to stabilise the charge redistribution as the reaction reaches the transition state[13]. Therefore, the role of protein dynamics in enzyme catalysis remains a controversial topic[10,11,14].

A solution to this problem was envisioned by determining heavy-enzyme kinetic isotope effects: the ratio of a reaction rate constant obtained with an unlabelled enzyme to the reaction rate constant obtained with a heavy isotope-labelled version of that enzyme[5,6]. The strategy was predicated on extending the Born-Oppenheimer approximation[15,16] to proteins, whereby labelling of enzymes with heavy isotopes would reduce local vibrational frequencies near the bond vibration timescale without interfering with the electrostatic properties of the system[5,6]. Applying this approach to selected enzymes where the chemical step could be isolated resulted most commonly in a normal heavy-enzyme kinetic isotope effect. In other words, the reaction with the isotope-labelled enzyme proceeded more slowly through the chemical step[5–7,17–20]. This was interpreted as evidence for coupling of fast protein dynamics to the chemical step, with a reduction in protein vibrational frequencies decreasing the probability of either crossing the transition-state energy barrier (purine nucleoside phosphorylase, HIV-1 protease, alanine racemase, lactate dehydrogenase)[5,6,18,20] or reaching the tunnelling-ready state (old yellow enzymes)[7,17], or yet, increasing recrossing of the transition-state dividing surface (bacterial dihydrofolate reductase)[21,22]. As an exception, alkaline phosphatase showed no evidence for coupling of protein motions to

[1]School of Biology, Biomedical Sciences Research Complex, University of St Andrews, St Andrews, UK. [2]EaStCHEM School of Chemistry, Biomedical Sciences Research Complex, University of St Andrews, St Andrews, UK. ✉e-mail: rgds@st-andrews.ac.uk

chemistry[23]. Curiously, this approach has not yet been reported for enzymes modulated by allosteric effectors, even though protein motions at various timescales in these systems are proposed to mediate communication between allosteric effector binding and the active-site response[2,3,24,25].

ATP phosphoribosyltransferase (ATPPRT) (EC 2.4.2.17), the enzyme responsible for the first and flux-controlling step of histidine biosynthesis[26,27], offers an opportunity to apply this approach to a complex allosteric system. ATPPRT catalyses the nucleophilic attack of N1 of ATP on C1 of 5-phospho-α-D-ribose 1-pyrophosphate (PRPP), displacing pyrophosphate (PP$_i$) to generate $N^1$-(5-phospho-β-D-ribosyl)-ATP (PRATP) in the presence of Mg$^{2+}$ (Fig. 1a), and is allosterically inhibited by histidine to shut down the pathway[26,28]. ATPPRT activity is also dependent on KCl.[29–33] ATPPRT is the focus of protein engineering efforts to optimize histidine biocatalytic production[34], and a promising target for novel antibiotic discovery against some pathogenic bacteria, including *Acinetobacter baumannii* and *Mycobacterium tuberculosis*[30,35,36]. Unlike hexameric long-form ATPPRTs, where one polypeptide chain harbours the catalytic and regulatory domains[37], hetero-octameric short-form ATPPRTs constitute a more complex allosteric system made up of catalytic (HisG$_S$) and regulatory (HisZ) proteins where two dimers of HisG$_S$ flank a tetramer of HisZ[27,29,38–41]. HisG$_S$ has low catalytic activity on its own and is insensitive to histidine[29,42]. HisZ, a histidyl-tRNA synthetase paralogue without any catalytic activity of its own, binds to HisG$_S$ to form the ATPPRT holoenzyme, which allosterically activates catalysis by HisG$_S$ in the absence of histidine[29,30,40,42].

HisZ also contains the histidine binding site and allosterically inhibits ATPPRT catalysis in the presence of histidine, playing a dual regulatory role[40,43]. Owing to their architectural versatility alongside their biomedical and biotechnological importance, ATPPRTs have been model systems to interrogate allostery, dynamics, and catalysis[2,8,30,32,38,40,44,45].

For *A. baumannii* ATPPRT, unique among other reported ATPPRTs due to its reaction proceeding via a rapid equilibrium random mechanism[30], steady-state and pre-steady-state kinetics studies point to chemistry as the rate-limiting step for nonactivated HisG$_S$ (henceforth referred to as *Ab*HisG$_S$). Allosteric activation by HisZ (*Ab*HisZ) disproportionately enhances the chemical step, making product release rate-limiting for the hetero-octameric holoenzyme (henceforth referred to as *Ab*ATPPRT)[46]. With *Ab*HisG$_S$, the lack of a burst of product formation suggested no step after chemistry is rate-limiting for the reaction. In agreement, replacement of Mg$^{2+}$ by Mn$^{2+}$, which leads to more efficient charge balance at the transition state of the related *Psychrobacter arcticus* ATPPRT, increased *Ab*HisG$_S$ steady-state catalytic constant ($k_{cat}$), as it does *P. arcticus* HisG$_S$ $k_{cat}$[2,45], suggesting chemistry is the rate-limiting step for the nonactivated enzyme form[46]. On the other hand, with *Ab*ATPPRT, a burst of product formation was inferred, although it was too fast at 25 °C to observe directly even with rapid kinetics, suggesting a step after chemistry is rate-limiting. This was corroborated by high solvent viscosity effects, which showed PRATP diffusion from the enzyme to be rate-determining for $k_{cat}$[46]. At 5 °C, the burst phase could finally be observed with *Ab*ATPPRT; moreover, the single-turnover rate constant ($k_{STO}$), which was much higher than $k_{cat}$,

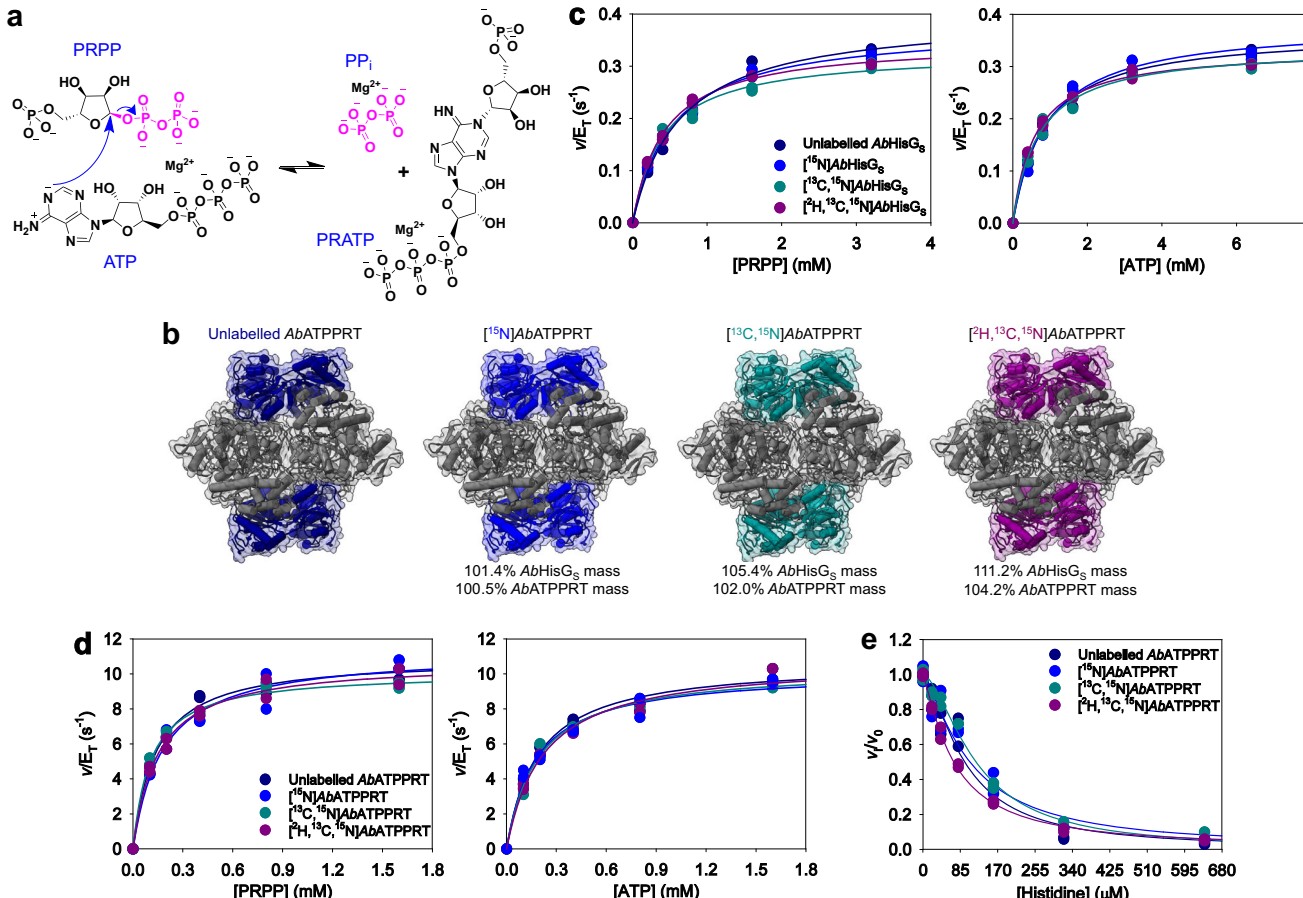

**Fig. 1 | The effects of isotope-labelling of *Ab*HisG$_S$ at 25 °C. a** The Mg$^{2+}$-dependent reversible reaction catalysed by ATPPRT. **b** Cartoon and surface representation of *Ab*ATPPRT (PDB ID 8OY0) with *Ab*HisZ (not isotope-labelled) in grey, and *Ab*HisG$_S$ colour-coded according to its isotope-labelling pattern. The mass increases are relative to the masses of *Ab*HisG$_S$ and *Ab*ATPPRT carrying natural isotope abundance. **c** Substrate saturation curves for *Ab*HisG$_S$ isotopologues. **d** Substrate saturation curves for *Ab*ATPPRT isotopologues. All data points are shown for two independent measurements. Lines are best fit of the data to Eq. (2). **e** Dose-response curves for histidine with *Ab*ATPPRT isotopologues in the presence of 1.4 mM ATP and 1.0 mM PRPP. All data points are shown. Two independent measurements were carried out, except for [$^{15}$N]*Ab*HisG$_S$ with 0–20 μM histidine, where three independent measurements were performed. Solid lines are best fits of the data to Eq. (5).

showed the chemical step is allosterically activated more than 1300-fold in $Ab$ATPPRT as compared with $Ab$HisG$_S$[46].

We hypothesize fast protein dynamics are involved in the significant allosteric enhancement of the chemical step in $Ab$ATPPRT, which would be susceptible to protein-mass modulation. Taking advantage of the fact that $Ab$HisG$_S$ and $Ab$HisZ are purified independently, and the $Ab$ATPPRT holoenzyme generated in vitro by mixing the two proteins at defined concentrations[30], we employed various isotope-labelling patterns of $Ab$HisG$_S$ accompanied by differential scanning fluorimetry (DSF), alternative-substrate kinetics, site-directed mutagenesis, steady-state and pre-steady-state enzyme kinetics, and temperature-rate profiles to probe the effect of increased protein mass on $Ab$ATPPRT catalysis and allostery.

## Results

### $Ab$HisG$_S$ and $Ab$ATPPRT activities are insensitive to protein mass at 25 °C

To assess the effect of increased $Ab$HisG$_S$ mass in catalysis, we purified $Ab$HisG$_S$ from heterologous expression in M9 medium supplemented with different isotopes to produce $Ab$HisG$_S$ carrying natural isotope abundance (unlabelled $Ab$HisG$_S$), [$^{15}$N]$Ab$HisG$_S$, [$^{13}$C,$^{15}$N]$Ab$HisG$_S$, and [$^2$H,$^{13}$C,$^{15}$N] $Ab$HisG$_S$ (the $^2$H is incorporated in non-exchangeable positions) (Supplementary Fig. 1). Electrospray ionisation/time-of-flight-mass spectrometry (ESI/TOF-MS) demonstrated the molecular masses of [$^{15}$N]$Ab$HisG$_S$, [$^{13}$C,$^{15}$N]$Ab$HisG$_S$, and [$^2$H,$^{13}$C,$^{15}$N]$Ab$HisG$_S$ increased by 1.4%, 5.4%, and 11.2%, respectively, from the unlabelled $Ab$HisG$_S$ molecular mass (Supplementary Fig. 2), which would result in increases of, respectively, 0.5%, 2.0%, and 4.2% in $Ab$ATPPRT masses, since the $Ab$HisZ mass[30] was never altered (Fig. 1b). DSF-based thermal denaturation assays showed $Ab$HisG$_S$ isotopologues display similar thermal unfolding profiles, and data fit to Eq. (1) yielded similar melting temperatures ($T_m$), except for [$^2$H,$^{13}$C,$^{15}$N] $Ab$HisG$_S$, which was ~4 °C less thermostable than its counterparts (Supplementary Fig. 3). $Ab$ATPPRT substrate saturation curves at 25 °C using unlabelled $Ab$HisG$_S$ produced from heterologous expression in LB medium by our published protocol[30] and unlabelled $Ab$HisG$_S$ generated here from expression in M9 medium and in M9 with high cell density induction[47] (the method used henceforth for all $Ab$HisG$_S$ produced in this work) showed negligible effects of different expression protocols and illustrated the expected data spread from one protein preparation to another (Supplementary Fig. 4; Supplementary Table 1).

Substrate saturation curves for $Ab$HisG$_S$ isotopologues at 25 °C (Fig. 1c) were fit to Eq. (2) to yield apparent steady-state kinetic parameters (Supplementary Table 2). While $k_{cat}$s for [$^{13}$C,$^{15}$N]$Ab$HisG$_S$ and [$^2$H,$^{13}$C,$^{15}$N] $Ab$HisG$_S$ are marginally lower than for unlabelled $Ab$HisG$_S$ and [$^{15}$N] $Ab$HisG$_S$, the overlapping nature of most data points in Fig. 1c disagrees with distinguishable heavy-enzyme kinetic isotope effects on $k_{cat}$ ($^{HE}k_{cat}$). As $Ab$HisG$_S$ $k_{cat}$ is limited by the chemical step, fast dynamics do not directly influence chemistry in this case. This was further probed by pre-steady-state kinetics under multiple-turnover conditions. The $Ab$HisG$_S$ reaction had been shown not to have a burst of product formation[46], which is reproduced here for unlabelled $Ab$HisG$_S$, whose traces overlap with those for [$^2$H,$^{13}$C,$^{15}$N]$Ab$HisG$_S$ (Supplementary Fig. 5).

Titration of $Ab$HisG$_S$ with $Ab$HisZ (Supplementary Fig. 6) showed activity increased for all isotopologues upon formation of the holoenzyme, and data fit to Eq. (3) resulted in the apparent equilibrium dissociation constants ($K_D$) for $Ab$HisZ in Supplementary Table 3, with no mass-dependent effects. This allows the concentrations of each $Ab$ATPPRT isotopologue to be calculated using Eq. (4). Substrate saturation curves for $Ab$ATPPRT isotopologues at 25 °C (Fig. 1d) were fit to Eq. (2), yielding apparent steady-state kinetic parameters (Supplementary Table 4). No $^{HE}k_{cat}$ was observed for $Ab$ATPPRT. This is not surprising as very high solvent viscosity effects on $Ab$ATPPRT $k_{cat}$ have shown this rate constant is determined by the diffusion of PRATP from the enzyme[46], which is not expected to depend on protein mass. Dose-response curves with histidine were best fit to Eq. (5) (Fig. 1e), yielding protein mass-independent half-maximal inhibitory concentrations (IC$_{50}$) and Hill coefficients ($n$).

(Supplementary Table 5) in range of previously reported values for unlabelled $Ab$ATPPRT[30].

### Allosteric activation triggers mass-dependent product release at 5 °C

Carrying out the $Ab$ATPPRT reaction at 5 °C permits observation of the rate of chemistry with rapid kinetics[46], a strategy we have repeated here to assess the effect of increased protein mass on the chemical step of $Ab$ATPPRT. Substrate saturation curves for $Ab$HisG$_S$ isotopologues showed no $^{HE}k_{cat}$ at 5 °C (Fig. 2a; Supplementary Table 6), which was corroborated by overlapping pre-steady-state kinetics traces for all $Ab$HisG$_S$ isotopologues under multiple-turnover conditions (Fig. 2b). These results confirm the lack of fast dynamics coupling to $Ab$HisG$_S$ chemistry at 5 °C, mirroring the results obtained at 25 °C. We have previously shown rapid kinetics of $Ab$ATPPRT under single-turnover conditions at 5 °C produces unimolecular single-turnover rate constants ($k_{STO}$) when $Ab$ATPPRT concentration is higher than 75 µM. In agreement with the presence of a burst of PRATP formation under multiple-turnover conditions, all $k_{STO}$ were much higher than $k_{cat}$, indicating that chemistry is fast in the holoenzyme[46]. We reproduced those results here with unlabelled $Ab$ATPPRT from two different batches and with [$^{13}$C,$^{15}$N]$Ab$ATPPRT and [$^2$H,$^{13}$C,$^{15}$N]$Ab$ATPPRT, all at 80 µM enzyme (Fig. 2c). The $Ab$HisZ $K_D$ at 5 °C was also determined for all $Ab$HisG$_S$ isotopologues (Supplementary Fig. 7; Supplementary Table 3). In the single-turnover kinetics, as previously reported[46], a short lag time in PRATP production is observed, and the data were best fit to Eq. (6), describing product formation in two consecutive irreversible steps: probably an isomerisation (e.g. a conformational change) of the $Ab$ATPPR-T:ATP:PRPP complex followed by on-enzyme formation of PRATP. There was no mass-dependence of the rate constants (Supplementary Table 7). This suggests the chemical step in $Ab$ATPPRT catalysis is not linked to fast protein motions.

Surprisingly, substrate saturation curves for $Ab$ATPPRT isotopologues showed a clear mass-dependence on $k_{cat}$ (Fig. 2d, Table 1), an effect that manifested itself only upon allosteric activation of the enzyme. As expected due to just a 0.5% increase in $Ab$ATPPRT mass, the [$^{15}$N]$Ab$ATPPRT $^{HE}k_{cat}$ ($^{15}k_{cat}$) is not statistically significant, but the $^{13,15}k_{cat}$ and $^{2,13,15}k_{cat}$ are. These results pose a conundrum, since a step after chemistry remains rate-limiting for $Ab$ATPPRT $k_{cat}$ at low temperature[46], but PRATP diffusional release, which is rate-limiting at 25 °C, is incompatible with protein-mass dependence. The $T_m$s are identical for unlabelled $Ab$ATPPRT, [$^{13}$C,$^{15}$N] $Ab$ATPPRT, and [$^2$H,$^{13}$C,$^{15}$N]$Ab$ATPPRT, and the presence of PRATP led only to marginal and mass-independent increases in $T_m$ (Supplementary Fig. 8; Supplementary Table 8).

We considered whether a change in rate-limiting step, still subsequent to chemistry, occurred at low temperature. To test this, $Ab$ATPPRT $k_{cat}$ at saturating concentrations of substrates was determined at 5 °C in the presence of increasing levels of the microviscogen glycerol (Fig. 2e). Determining the $K_D$ for $Ab$HisZ at 12% glycerol (Supplementary Fig. 9) demonstrated that $Ab$HisG$_S$ remained saturated with $Ab$HisZ in the presence of the microviscogen. Furthermore, increasing the concentration of $M.$ tuberculosis pyrophosphatase ($Mt$PPase) did not alter the $Ab$ATPPRT rate (Supplementary Fig. 10), indicating the rate remains independent of the coupled enzyme concentration at 12% glycerol. We use $Mt$PPase in ATPPRT assays to drive forward the reaction equilibrium, rendering the reaction essentially irreversible[29]. A plot of $k_{cat}$ ratios against relative viscosity produced a slope of 0.03 ± 0.06 upon best fit of the data to Eq. (7), indistinguishable from 0 within experimental error, indicating diffusional steps do not contribute to $Ab$ATPPRT $k_{cat}$ at 5 °C, in sharp contrast to the scenario at 25 °C, where a similar analysis had yielded a slope of 0.96 ± 0.07[46], within experimental error of the theoretical maximum value of 1, indicating rate-determining diffusion[48].

A kinetic sequence is proposed to describe the kinetically relevant steps encompassed by $Ab$ATPPRT $k_{cat}$ at 5 °C (Fig. 2f). Reproducing what we previously observed, an isomerisation of the Michaelis complex is followed by the chemical step, here including the fast release of PP$_i$ which is

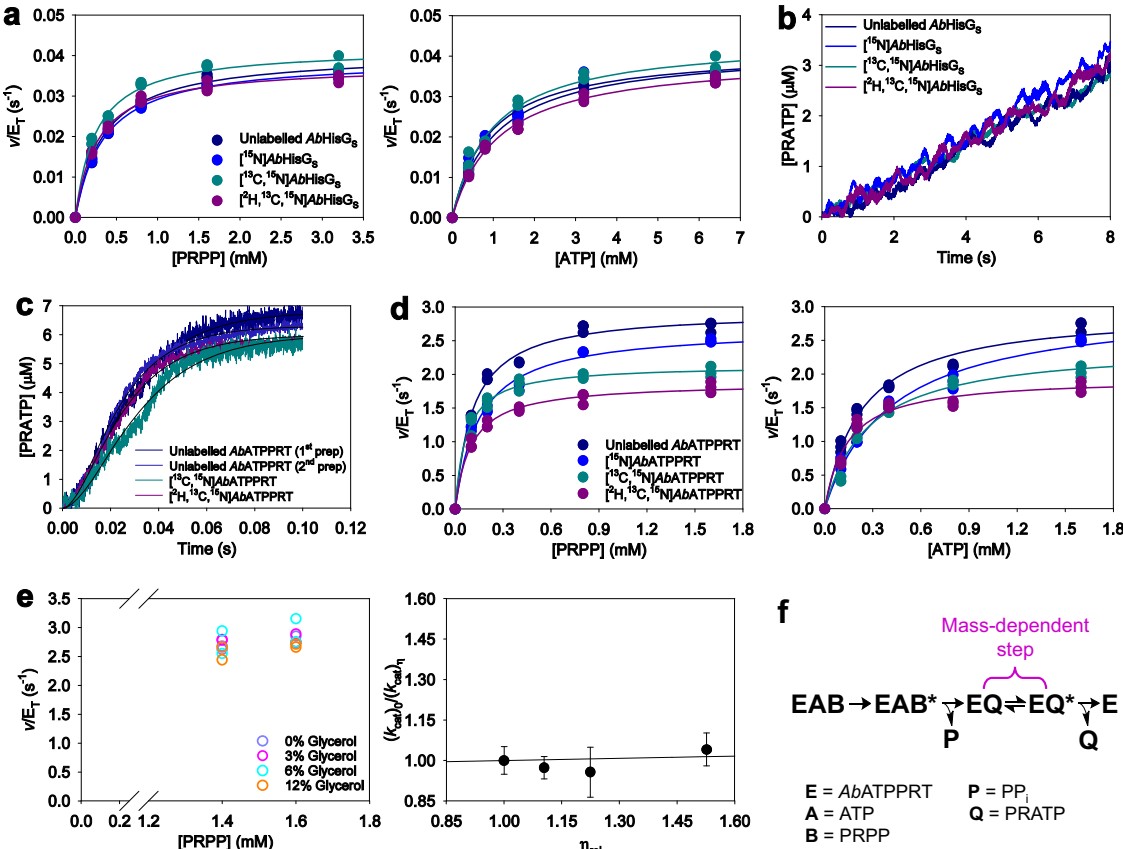

**Fig. 2 | The effects of isotope-labelling of $Ab$HisG$_S$ at 5 °C. a** Substrate saturation curves for unlabelled and isotope-labelled $Ab$HisG$_S$. All data points for two independent measurements are shown. Lines are best fit of the data to Eq. (2). **b** Rapid kinetics of PRATP formation at 5 °C by unlabelled $Ab$HisG$_S$ and isotope-labelled $Ab$HisG$_S$ under multiple-turnover conditions. Lines are averages of fifteen replicates. **c** Pre-steady-state kinetics of PRATP formation by $Ab$ATPPRT isotopologues under single-turnover conditions. Lines in colour are averages of six replicates. Thin black lines are best fit of the data to Eq. (6). **d** Substrate saturation curves for $Ab$ATPPRT isotopologues. All data points for two independent measurements are shown. Lines are best fit of the data to Eq. (2). **e** Solvent viscosity effects on unlabelled $Ab$ATPPRT $k_{cat}$. All data points for two independent measurements at each PRPP concentration are shown as open circles. Closed circles are the mean ± SD of four measurements at all PRPP concentrations. The line is best fit of the data to Eq. (7). **f** Kinetic sequence encompassed by $Ab$ATPPRT $k_{cat}$ at 5 °C, highlighting the step proposed to depend on protein mass. The asterisk denotes an isomerised complex.

**Table 1 | Apparent steady-state kinetic parameters and $^{HE}k_{cat}$ (mean ± fitting error) at 5 °C for $Ab$ATPPRT isotopologues**

| $Ab$ATPPRT isotopologue | $K_M^{PRPP}$ (mM) | $K_M^{ATP}$ (mM) | $k_{cat}$ (s$^{-1}$) | $k_{cat}/K_M^{PRPP}$ (M$^{-1}$ s$^{-1}$) | $k_{cat}/K_M^{ATP}$ (M$^{-1}$ s$^{-1}$) | $^{HE}k_{cat}$ |
|---|---|---|---|---|---|---|
| Unlabelled | 0.111 ± 0.009 | 0.23 ± 0.02 | 2.95 ± 0.03 | (2.7 ± 0.2) × 10$^4$ | (1.3 ± 0.1) × 10$^4$ | 1.00 ± 0.01 |
| $^{15}$N | 0.15 ± 0.02 | 0.41 ± 0.05 | 2.87 ± 0.06 | (1.9 ± 0.3) × 10$^4$ | (7.0 ± 0.9) × 10$^3$ | 1.03 ± 0.02 |
| $^{13}$C,$^{15}$N | 0.069 ± 0.008 | 0.27 ± 0.04 | 2.27 ± 0.06* | (3.3 ± 0.4) × 10$^4$ | (8 ± 1) × 10$^3$ | 1.30 ± 0.04 |
| $^2$H,$^{15}$N | 0.085 ± 0.008 | 0.16 ± 0.01 | 2.05 ± 0.03* | (2.4 ± 0.2) × 10$^4$ | (1.28 ± 0.08) × 10$^4$ | 1.44 ± 0.03 |
| $^2$H,$^{13}$C,$^{15}$N | 0.099 ± 0.009 | 0.14 ± 0.02 | 1.92 ± 0.04* | (1.9 ± 0.2) × 10$^4$ | (1.4 ± 0.2) × 10$^4$ | 1.54 ± 0.04 |

*$p < 0.01$, by a Student's $t$-test in comparison with unlabelled $Ab$ATPPRT $k_{cat}$.

immediately hydrolysed by $Mt$PPase, making the reaction irreversible[46]. These steps are insensitive to protein mass. Absence of solvent viscosity effects suggests the $Ab$ATPPRT:PRATP complex undergoes a rate-limiting isomerisation preceding PRATP departure. As this step is mass-dependent, this isomerisation likely involves a protein vibrational motion. At 25 °C, such motion is fast, and PRATP release becomes diffusion-limited, masking the $^{HE}k_{cat}$.

**"Stress testing" the $Ab$ATPPRT heavy-enzyme isotope effects**
To challenge further the unusual $^{HE}k_{cat}$ reporting on PRATP release from $Ab$ATPPRT, an additional $Ab$HisG$_S$ isotopologue was generated, [$^2$H,$^{15}$N] $Ab$HisG$_S$ (Fig. 3a; Supplementary Fig. 1e), resulting in increases of 7.0% and

2.6% over the molecular masses of unlabelled $Ab$HisG$_S$ and $Ab$ATPPRT, respectively (Supplementary Fig. 2e). The $Ab$HisZ $K_D$s at 5 and 25 °C were also determined for [$^2$H,$^{15}$N]$Ab$HisG$_S$ (Supplementary Fig. 11; Supplementary Table 3). Substrate saturation curves for [$^2$H,$^{15}$N]$Ab$ATPPRT at 25 °C again showed no mass-dependent effect (Fig. 3b; Supplementary Table 4), but at 5 °C, the [$^2$H,$^{15}$N]$Ab$ATPPRT saturation curves fell qualitatively between those of [$^{13}$C,$^{15}$N]$Ab$ATPPRT and [$^2$H,$^{13}$C,$^{15}$N]$Ab$ATPPRT (Fig. 3c), although the [$^2$H,$^{15}$N]$Ab$ATPPRT $k_{cat}$ was not statistically distinct from either the [$^{13}$C,$^{15}$N]$Ab$ATPPRT or the [$^2$H,$^{13}$C,$^{15}$N]$Ab$ATPPRT $k_{cat}$ (Table 1).

To dissect the influence temperature exerts on the protein mass-dependence of $Ab$ATPPRT $k_{cat}$, the $^{HE}k_{cat}$ variation with protein mass was

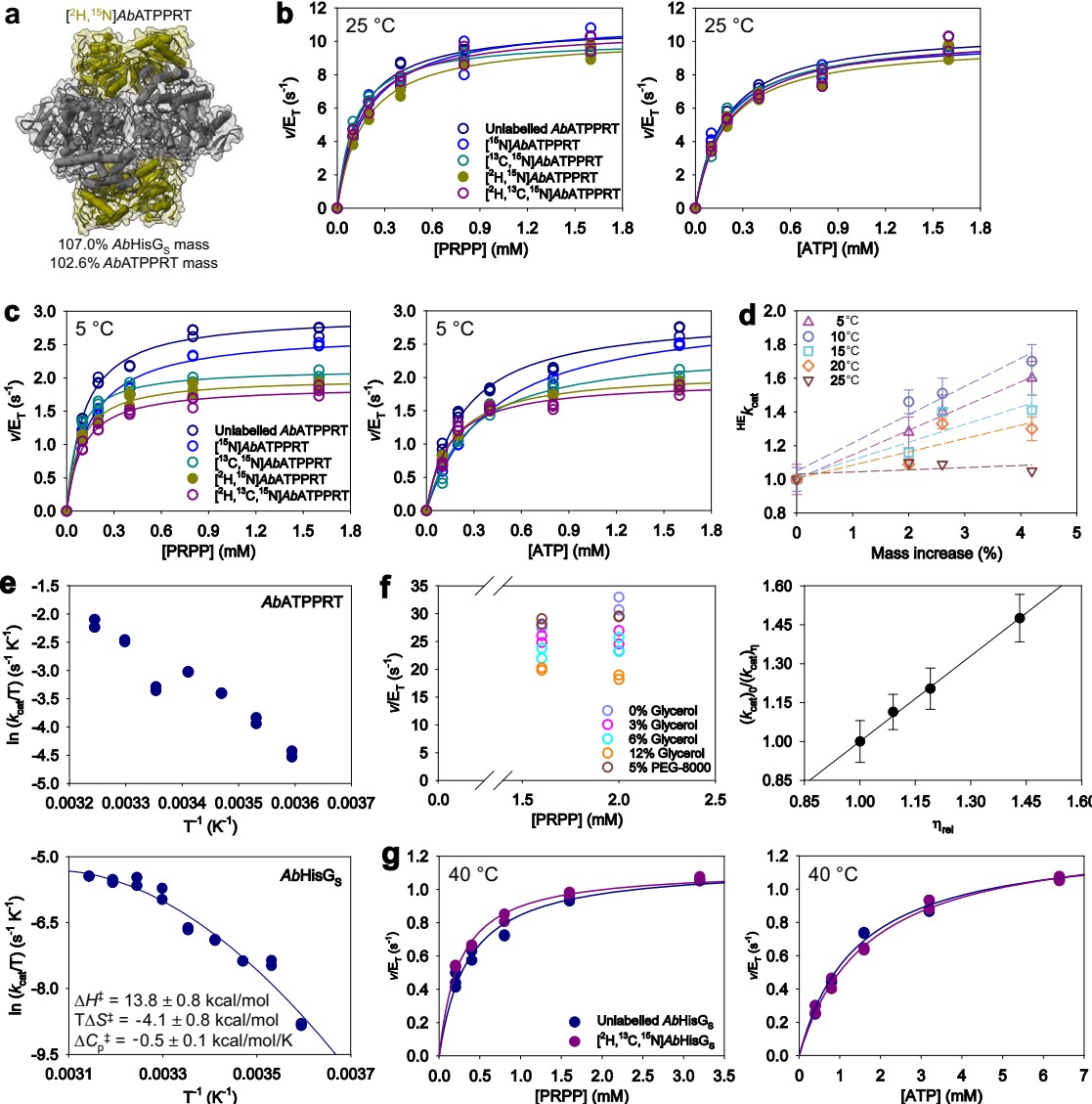

**Fig. 3 | Additional mass and temperature probes of AbATPPRT $^{HE}k_{cat}$. a** Cartoon and surface representation of AbATPPRT (PDB ID 8OY0) with AbHisZ (never isotope-labelled) in grey, and [²H,¹⁵N]AbHisGₛ in yellow. The mass increases are relative to the masses of AbHisGₛ and AbATPPRT carrying natural isotope abundance. **b** Substrate saturation curves for AbATPPRT isotopologues at 25 °C. All data points for two independent measurements are shown. Lines are best fit of the data to Eq. (2). Open circles denote the same data first depicted in Fig. 1d. **c** Substrate saturation curves for AbATPPRT isotopologues at 5 °C. All data points for two independent measurements are shown. Lines are best fit of the data to Eq. (2). Open circles denote the same data first depicted in Fig. 2d. **d** Temperature-dependence of the ¹³,¹⁵$k_{cat}$, ²,¹⁵$k_{cat}$, and ²,¹³,¹⁵$k_{cat}$. Data are shown as mean ± SD of three independent measurements, except at 25 °C where they are the mean of two independent measurements. Dashed lines are linear regressions of the data intended only to aid the eye. **e** Temperature profiles of AbATPPRT (278–308 K) and AbHisGₛ (278–318 K) $k_{cat}$s. All data points of two independent measurements are shown. The line is the best fit to Eq. (8), which produced the activation thermodynamics parameters (mean ± fitting error) shown as inset. The $\Delta S^{\ddagger}$ and $\Delta H^{\ddagger}$ are those at 25 °C (298 K). **f** Solvent viscosity effects on AbATPPRT $k_{cat}$ at 35 °C. All data points for two independent measurements at each PRPP concentration are shown as open circles, except at 0% glycerol (v/v) where three independent measurements were carried out. Closed circles are the mean ± SD of four measurements at all PRPP concentrations, except at 0% glycerol (v/v), with six measurements at all PRPP concentrations. The line is best fit of the data to Eq. (7). **g** Substrate saturation curves for unlabelled and [²H,¹³C,¹⁵N]AbHisGₛ at 40 °C (318 K). All data points for two independent measurements are shown. Lines are best fit of the data to Eq. (2).

evaluated at increasing temperatures from 5 to 25 °C (Supplementary Fig. 12). While the data trended to more pronounced $^{HE}k_{cat}$ at low temperatures, the relationship was not strict (Fig. 3d). The $^{HE}k_{cat}$s increased monotonically with the relative increase in AbATPPRT mass at 5 and 10 °C, but were higher at 10 °C. The $^{HE}k_{cat}$s decreased at 15 °C and again at 20 °C, but their dependence on protein mass became more disperse. Finally, at 25 °C the $^{HE}k_{cat}$ collapsed, probably masked by the AbATPPRT $k_{cat}$ becoming fully determined by a diffusional step instead of a protein vibration.

The data displayed in Supplementary Fig. 12 indicate that AbATPPRT $k_{cat}$ is lower at 25 °C than at 20 °C. To confirm and explore this unusual

result, we characterised the temperature-rate profile of AbATPPRT and AbHisGₛ (Fig. 3e). For the AbATPPRT profile, the $K_D$ for AbHisZ was determined at 35 °C (Supplementary Fig. 13), to ensure the AbHisGₛ remained saturated with AbHisZ even at the highest temperature used. Temperature stability tests showed that AbATPPRT is not stable to incubation for 10 min at 40 °C, upon which activity at 30 °C drops precipitously (Supplementary Fig. 14), even though the AbATPPRT $T_m$ is ~55 °C. As ATPPRT catalysis is dependent on KCl[29–33], the rate-dependence on KCl concentration was evaluated at 25 and 35 °C (Supplementary Fig. 15), demonstrating that maximum activity is achieved with 100 mM KCl at both temperatures. AbATPPRT activity was determined at saturating

concentration of both substrates at every temperature (Supplementary Fig. 16), ensuring the maximum rate was achieved. The $Ab$ATPPRT temperature-rate profile determined between 5 and 35 °C (278 and 308 K) indeed confirmed the reduction in $k_{cat}$ at 25 °C from that at 20 °C, followed again by a rise in $k_{cat}$ between 25 and 35 °C (Fig. 3e). The most likely explanation for this unusual behaviour is the change in rate-limiting step from a protein vibration-limited product release below 20 °C to a diffusion-limited product release from 25 °C onwards. At 35 °C, $k_{cat}$ decreases as the glycerol concentration increases, but is unchanged in the presence of the macroviscogen polyethylene glycol-8000 (PEG-8000), demonstrating sizable viscosity effects on PRATP release due to increased microviscogen concentration (Fig. 3f). The $K_D$ for $Ab$HisZ determined at 12% glycerol at 35 °C (Supplementary Fig. 17) indicates $Ab$HisG$_S$ remained saturated with $Ab$HisZ in the presence of glycerol at the 35 °C. The rate was unaltered upon increasing $Mt$PPase concentration (Supplementary Fig. 18), indicating the reaction remains coupled at the highest temperature employed. A plot of $k_{cat}$ ratios against relative viscosity produced a slope of 1.09 ± 0.05 (Fig. 3f), indicating a diffusional step determines $k_{cat}$ at 35 °C as it does at 25 °C[46].

The $Ab$HisG$_S$ temperature-rate profile was determined between 5 and 45 °C (278 and 318 K) (Fig. 3e). The rate-dependence on KCl concentration showed maximum activity is achieved with 100 mM KCl at 25 and 40 °C (Supplementary Fig. 15), and both ATP and PRPP concentrations were saturating at all temperatures (Supplementary Fig. 19). Intriguingly, the Eyring plot was nonlinear (Fig. 3e). Substrate saturation curves at 40 °C showed $k_{cat}$ is enhanced when Mn$^{2+}$ replaces Mg$^{2+}$ as the divalent metal (Supplementary Fig. 19), as observed at 5 and 25 °C[46], suggesting the same rate-limiting step (likely chemistry) is operational at high temperature. This raises the possibility $Ab$HisG$_S$ catalysis involves nonzero heat capacity of activation ($\Delta C_P^{\ddagger}$), and fitting the Eyring plot to Eq. (8) produced thermodynamic parameters of activation shown in Fig. 3e. A negative $\Delta C_P^{\ddagger}$ on $k_{cat}$ predicts a reduction in the number of vibrational modes available to absorb energy in the transition state as compared with the ground state (Michaelis complex)[49]. The $Ab$HisG$_S$ $k_{cat}$ responded less and less to increases in temperature beyond 30 °C (Fig. 3e). We thus conjectured that any coupling of non-statistical motions on chemical barrier crossing might be brought to the fore as the $\Delta G^{\ddagger}$ term becomes less responsive to temperature. We evaluated the effect of reducing protein vibrational frequencies on the reaction rate at 40 °C (Fig. 3g). No $^{HE}k_{cat}$ was obtained, indicating no enhanced coupling of fast protein vibrations to transition-state barrier crossing in $Ab$HisG$_S$ at high temperature.

## Replacement of ATP by ADP abolishes the $Ab$ATPPRT heavy-enzyme isotope effects

Crystal structures of *P. arcticus* ATPPRT in complexes with PRPP:ATP and with PRPP:ADP (Fig. 4a) revealed remarkably similar Michaelis complexes, except for a salt-bridge between ATP γ-PO$_4$$^{2-}$ in one HisG$_S$ subunit and

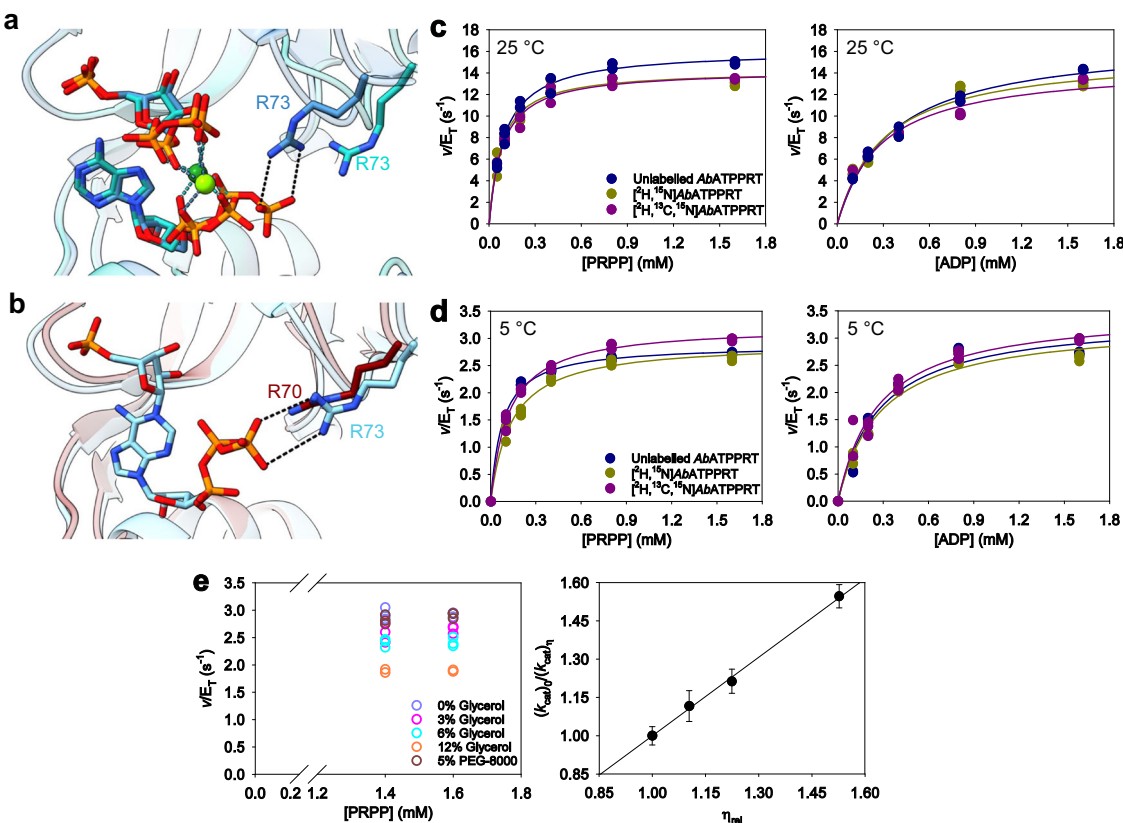

**Fig. 4 | The effect of ADP as substrate on $Ab$ATPPRT $^{HE}k_{cat}$. a** Ribbon diagram of overlaid active sites of *P. arcticus* ATPPRT in complexes with PRPP:ATP (PDB ID 6FU2) and PRPP:ADP (PDB ID 6FUA), with carbon atoms in royal blue and cyan, respectively. Only the interaction between ATP and Arg73 is shown (dashed lines). **b** Ribbon diagram of overlaid active sites of *P. arcticus* ATPPRT in complex with PRATP (PDB ID 6FU7) and unliganded $Ab$ATPPRT (PDB ID 8OY0), with carbon atoms in cyan and maroon, respectively. The only interaction shown as dashed lines is that between PRATP and Arg73 (the equivalent Arg70 in $Ab$ATPPRT is shown in maroon). **c** Substrate saturation curves for $Ab$ATPPRT isotopologues at 25 °C with ADP as substrate. All data points for two independent measurements are shown, except: 0.1 mM PRPP/unlabelled $Ab$ATPPRT, 0.05 mM PRPP/[$^2$H,$^{15}$N]$Ab$ATPPRT,

0.4 mM PRPP/[$^2$H,$^{13}$C,$^{15}$N]$Ab$ATPPRT (varying PRPP), and 0.8 mM PRPP/ [$^2$H,$^{13}$C,$^{15}$N]$Ab$ATPPRT (varying ADP), where three independent measurements were carried out. Lines are best fit of the data to Eq. (2). **d** Substrate saturation curves for $Ab$ATPPRT isotopologues at 5 °C with ADP as substrate. All data points for three independent measurements are shown. Lines are best fit of the data to Eq. (2). **e** Solvent viscosity effects on $Ab$ATPPRT $k_{cat}$ at 5 °C with ADP as substrate. All data points for three independent measurements at each PRPP concentration are shown as open circles, except at 12% glycerol (v/v) where two independent measurements were carried out. Closed circles are the mean ± SD of six measurements at all PRPP concentrations, except at 12% glycerol (v/v), with four measurements at all PRPP concentrations. The line is best fit of the data to Eq. (7).

Arg73 of the adjacent HisG$_S$ subunit, which is missing in the PRPP:ADP complex[38]. This interaction is also present in the *P. arcticus* ATPPRT:-PRATP binary complex[38], and overlay of this structure with that of unli-ganded *Ab*ATPPRT[46] (Fig. 4b) illustrates the equivalent Arg70 of *Ab*HisG$_S$ might make a similar interaction in a complex with PRATP. Since product release is rate-limiting for *Ab*ATPPRT $k_{cat}$, whereas chemistry is rate-limiting for *Ab*HisG$_S$ $k_{cat}$, the presumed absence of this interaction when $N^1$-(5-phospho-β-D-ribosyl)-ADP (PRADP) is the product may contribute to the higher *Ab*ATPPRT $k_{cat}$, but unaltered *Ab*HisG$_S$ $k_{cat}$, obtained when ADP is the substrate[46]. To gain insight into the *Ab*ATPPRT $^{HE}k_{cat}$ at the molecular level, we attempted to perturb the $^{HE}k_{cat}$ obtained with [$^2$H,$^{15}$N] *Ab*ATPPRT and [$^2$H,$^{13}$C,$^{15}$N]*Ab*ATPPRT by replacing ATP with ADP. No statistically significant mass-dependence of *Ab*ATPPRT $k_{cat}$ was seen at 25 °C (Fig. 4c, Supplementary Table 9). Interestingly, the unlabelled *Ab*ATPPRT $k_{cat}$ with ADP at 5 °C (Supplementary Table 10) is the same as that with ATP (Table 1), but the use of ADP abolished the *Ab*ATPPRT $^{HE}k_{cat}$ (Fig. 4d, Supplementary Table 10). The presence of $^{HE}k_{cat}$ with ATP but not with ADP as substrate suggests the PRATP γ-PO$_4$$^{2-}$-Arg70 interaction might be involved in the mass-dependence of the *Ab*ATPPRT $k_{cat}$. How-ever, the similar *Ab*ATPPRT $k_{cat}$ with either nucleotide would point to the same step limiting both reactions. Surprisingly, contrary to what was obtained with ATP (Fig. 2e), high solvent viscosity effects at 5 °C with ADP as substrate were observed, and the plot of $k_{cat}$ ratios against relative viscosity produced a slope of 1.03 ± 0.02 (Fig. 4e), compatible with PRADP diffusion from the enzyme determining $k_{cat}$ at 5 °C, despite the unaltered *Ab*ATPPRT $k_{cat}$ compared with that reflecting a protein motion underpinning release of PRATP.

### *Ab*HisG$_S$ Arg70 is implicated in the *Ab*ATPPRT $^{HE}k_{cat}$

Arg70 is conserved in 67% of the 205 HisG$_S$ amino-acid sequences used in a multiple sequence alignment, a snapshot of which, containing the 20 most similar sequences to *Ab*HisG$_S$ is shown in Supplementary Fig. 21. The other 33% of sequences harbour a Lys residue in the equivalent position. Mapping the sequence conservation onto the structure of the *Ab*HisG$_S$ dimer (Fig. 5a) produced a structural picture of the most and least conserved regions of the protein, based on a conservation index[50]. Arg70 is the most conserved residue in a three-residue loop, flanked by Leu69 (very low conservation) and Ala71 (moderate conservation) (Fig. 5b).

To check the importance of Arg70 in *Ab*ATPPRT catalysis and uncover clues about the protein mass-dependent isotope effects at the molecular level, we produced an R70A-*Ab*HisG$_S$ variant and its iso-topologue [$^{13}$C,$^{15}$N]R70A-*Ab*HisG$_S$ (Supplementary Fig. 22). Intriguingly, the mutation increased the $T_m$ of both unlabelled R70A-*Ab*HisG$_S$ and [$^{13}$C,$^{15}$N]R70A-*Ab*HisG$_S$ by ~5 °C from the corresponding WT-*Ab*HisG$_S$ $T_m$, but binding to *Ab*HisZ restored the $T_m$s to their WT-*Ab*ATPPRT values (Supplementary Fig. 23). The R70A-*Ab*HisG$_S$ rates at 25 °C at substrate concentrations that would be saturating for WT-*Ab*HisG$_S$ (Supplementary Fig. 24) were significantly reduced ~131-fold compared to WT-*Ab*HisG$_S$ $k_{cat}$. Both R70A-*Ab*HisG$_S$ and [$^{13}$C,$^{15}$N]R70A-*Ab*HisG$_S$ are allosterically activated by *Ab*HisZ without detriment to *Ab*HisZ affinity both at 5 °C and 25 °C (Supplementary Fig. 25; Supplementary Fig. 26). The R70A mutation caused an ~5-fold decrease in $k_{cat}$ compared to WT-*Ab*ATPPRT, and nullified the higher *Ab*ATPPRT $k_{cat}$ with ADP as substrate observed with the WT variant at 25 °C (Fig. 5c; Supplementary Table 11). These results agree with a role for Arg70 in PRATP release, as predicted above.

Since no statistically significant $^{HE}k_{cat}$ had been observed with ADP as substrate, we hypothesized the R70A mutation would abrogate the *Ab*ATPPRT $^{HE}k_{cat}$ even with ATP as substrate because the Arg70 interaction with the PRATP γ-PO$_4$$^{2-}$ would be disrupted. No statistically significant R70A-*Ab*ATPPRT $^{HE}k_{cat}$ was observed either at 5 °C (Fig. 5d; Supplemen-tary Table 12) or at 25 °C (Fig. 5e; Supplementary Table 13). To assess the effect of the R70A mutation on the mechanism of PRATP release at 25 °C, solvent viscosity effects were measured on R70A-*Ab*ATPPRT $k_{cat}$ (Fig. 5f; Supplementary Fig. 27), and the plot of $k_{cat}$ ratios against relative viscosity produced a slope of 0.06 ± 0.05, suggesting diffusional steps are kinetically

silent, in sharp contrast with the WT-*Ab*ATPPRT where PRATP diffusion determines $k_{cat}$[46]. This result also reveals that the R70A-*Ab*ATPPRT protein mass-dependent isotope effects can be obscured even in the absence of a diffusion-limited step.

The lack of solvent viscosity effects raised the possibility that R70A-*Ab*ATPPRT $k_{cat}$ is limited by chemistry. Unlike the WT enzyme, the R70A-*Ab*ATPPRT $k_{STO}$ could be measured at 25 °C (Supplementary Fig. 28), and data fit to Eq. (9) yielded a value of 36 ± 2 s$^{-1}$ (mean ± SD) with 80 μM unlabelled R70A-*Ab*ATPPRT. The $k_{STO}$ increased to 44 ± 2 s$^{-1}$ with 100 μM unlabelled R70A-*Ab*ATPPRT (Fig. 5g), whose average trace overlapped with that obtained with 100 μM [$^{13}$C,$^{15}$N]R70A-*Ab*ATPPRT ($k_{STO}$ of 42 ± 1 s$^{-1}$). These observations indicate the measured $k_{STO}$ are not unim-olecular and probably have a contribution from binding, since increasing protein concentration led to an increase in $k_{STO}$. Regardless, chemistry is at least 20-fold faster than a subsequent step and therefore does not limit R70A-*Ab*ATPPRT $k_{cat}$. Finally, the lack of heavy-enzyme kinetic isotope effect on $k_{STO}$ does not support coupling of fast protein vibrations to the chemical step in R70A-*Ab*ATPPRT catalysis.

## Discussion

Previous studies have reported isotope-labelling of the protein decreased $k_{cat}$ in systems where this macroscopic rate constant is limited by product release instead of chemistry (dihydrofolate reductase, formate dehy-drogenase, methylthioadenosine nucleosidase) and takes place at slower timescales (μs–ms)[22,51–54]. However, these effects differ from the ones observed for *Ab*ATPPRT. At least for formate dehydrogenase and a cold-adapted dihydrofolate reductase, where double- and triple-isotope-labelling patterns produced two different molecular masses for the labelled enzymes, the effect on $k_{cat}$ was not mass-dependent, i.e. isotope-labelled enzymes with distinct masses had the same $k_{cat}$ reduction[22,53]. Mass modulation of such low-frequency motions are not readily intuitive, since the typically small increase in protein mass (e.g. 5–11%) would lead to an imperceptible decrease in vibration frequency due to negligibly low restoring force constants[53,55]. The results have instead been attributed to the breakdown of the Born-Oppenheimer approximation and a con-sequent perturbation of the electrostatic potential surface of the isotope-labelled proteins[53,55].

Protein mass-dependent effects on *Ab*ATPPRT are unique because the heavy-enzyme kinetic isotope effects are evident exclusively upon allosteric activation of the enzyme, are strictly protein-mass-dependent, but are only manifested on $k_{cat}$, which in *Ab*ATPPRT ultimately emerges as millisecond-timescale events. The $^{HE}k_{cat}$ are attributed to a rate-limiting protein motion associated with PRATP release below 25 °C, above which this motion pre-sumably speeds up and PRATP release becomes diffusional and insensitive to protein mass.

Intriguingly, the absence of *Ab*ATPPRT $^{HE}k_{STO}$ and *Ab*HisG$_S$ $^{HE}k_{cat}$ speak against a coupling of fast protein vibrations to the chemical step of catalysis in allosterically activated and nonactivated forms of this enzyme, with the important caveat that *Ab*ATPPRT chemistry is too fast near physiological temperature to capture even with stopped-flow spectrophotometry[46], so that no mass-dependent effect could be eval-uated. The only other report of lack of heavy-enzyme isotope effects on the chemical step is the metalloenzyme alkaline phosphatase. In that instance, the unaltered mass of the catalytic Zn$^{2+}$ ions, involved in water activation and leaving group departure stabilisation, was invoked as a potential explanation for the decoupling between protein mass and chemical barrier crossing[23]. While ATPPRT is not per se a metalloenzyme, the reaction is dependent on Mg$^{2+}$ ions[26], and crystal structures, computational chemistry calculations, and enzyme kinetics suggest it is involved directly in chemistry by positioning the substrates for nucleophilic attack and by stabilising the pyrophosphate leaving group[2,38,45]. One might hypothesise, by analogy to the alkaline phosphatase proposal, the unaltered mass of Mg$^{2+}$ prevents cou-pling from fast protein motions to chemistry in *Ab*ATPPRT. Expanding these studies to other ATPPRTs and other metalloenzymes may provide a test of this hypothesis.

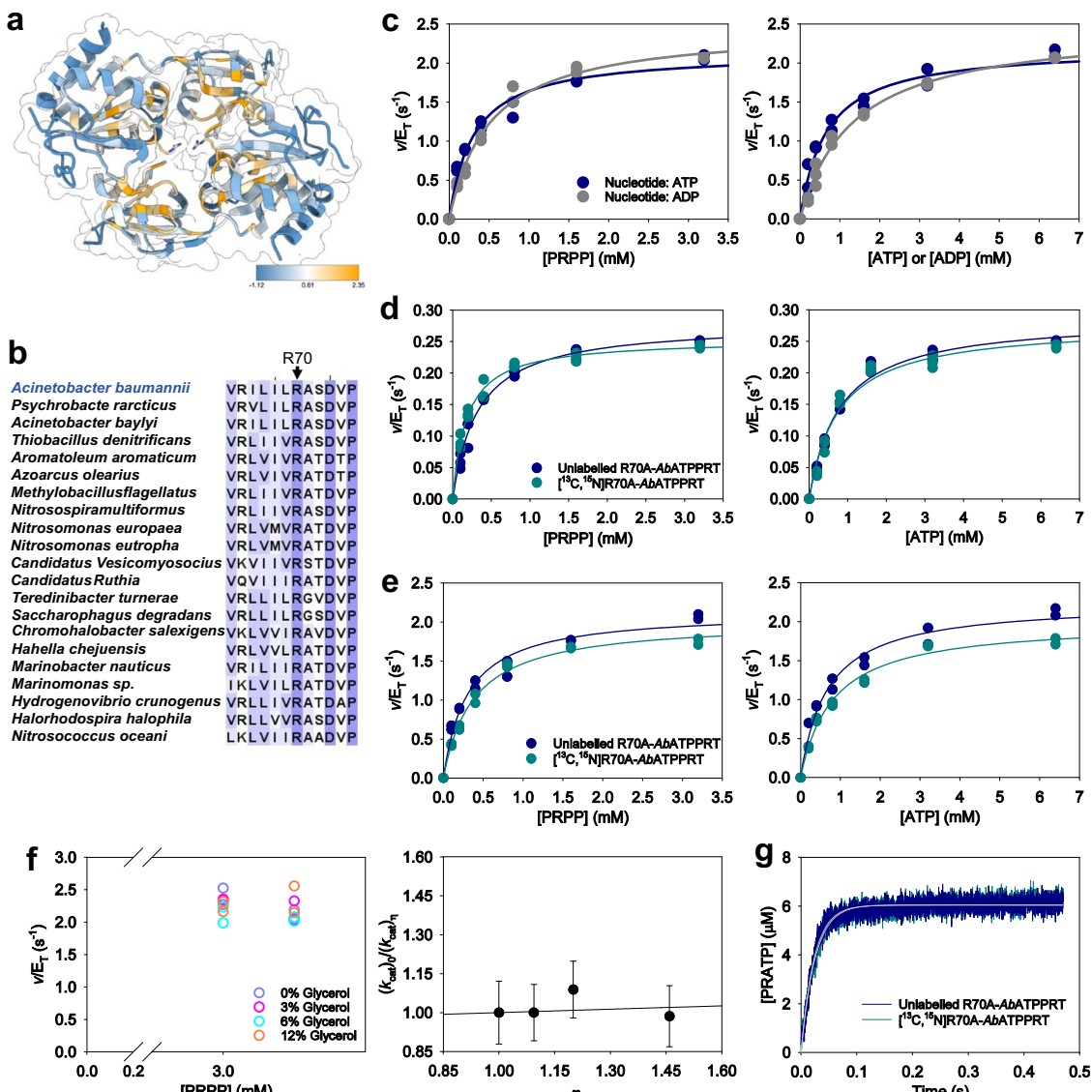

**Fig. 5 | The effect of R70A mutation on *Ab*ATPPRT. a** Amino-acid sequence conservation mapped onto the *Ab*HisG$_S$ dimer structure (PDB ID 8OY0) from highly conserved (score 2.35) to not conserved (score −1.12). Arg70 are shown in stick model. **b** Close-up view of the amino-acid sequence alignment depicted in Supplementary Fig. 21, flanking the Arg70 position. **c** Substrate saturation curves for unlabelled R70A-*Ab*ATPPRT at 25 °C with either ATP or ADP as substrate. All data points for two independent measurements are shown, except 0.4 mM ADP, where three data points are shown. Lines are best fit of the data to Eq. (2). **d** Substrate saturation curves for R70A-*Ab*ATPPRT isotopologues at 5 °C. All data points for three independent measurements are shown. Lines are best fit of the data to Eq. (2).

**e** Substrate saturation curves for R70A-*Ab*ATPPRT isotopologues at 25 °C. All data points for two independent measurements are shown. Lines are best fit of the data to Eq. (2). **f** Solvent viscosity effects on R70A-*Ab*ATPPRT $k_{cat}$ at 25 °C. All data points for two independent measurements at each PRPP concentration are shown as open circles. Closed circles are the mean ± SD of four measurements at all PRPP concentrations. The line is best fit of the data to Eq. (7). **g** Pre-steady-state kinetics of PRATP formation by R70A-*Ab*ATPPRT isotopologues under single-turnover conditions at 25 °C. Lines in colour are averages of six replicates. Thin grey lines are best fit of the data to Eq. (9).

It is unlikely that increased *Ab*ATPPRT mass by isotope-labelling exerts its effect directly on a putative millisecond-timescale motion (*Ab*ATPPRT $k_{cat}$ is ~10 s$^{-1}$), as large-amplitude motions, common at this timescale, would not respond to very small mass changes[53,55]. Two non-mutually exclusive models might be invoked to account for the mass-dependent effects on *Ab*ATPPRT. One model relies on picosecond-timescale dynamics of backbone amide groups enabling global, microsecond-to-millisecond-timescale protein motions triggering product release; this is observed in adenylate kinase[56]. These picosecond fluctuations could be directly slowed down by isotope-labelling, and this mechanism cannot be ruled out as the origin of the *Ab*ATPPRT heavy-enzyme isotope effects. In adenylate kinase, these fast backbone motions are located at conserved hinge regions that control a large lid-opening motion to release

products[56]. While there are hotspots of amino-acid sequence conservation outside the active site in HisG$_S$ (Fig. 5a), their involvement in enabling PRATP release is yet unknown.

Another model is gleaned from the use of the non-physiological substrate ADP, and site-directed mutagenesis of *Ab*HisG$_S$. Disrupting a salt-bridge between Arg70 and PRATP γ-PO$_4$$^{2-}$ eliminates the $^{HE}k_{cat}$ (while chemistry remains fast and protein-mass insensitive), suggesting this interaction is important to bring about the $^{HE}k_{cat}$. Picosecond-timescale side-chain motions are common in proteins[57–59], and the long side chain of arginine residues, whose dynamics affects their polar and nonpolar interactions with ligands[60,61], can retain significant residual conformational entropy even while participating in salt-bridges[59]. Therefore, it is possible the Arg70 side-chain fast dynamics and consequently its interaction with

PRATP $\gamma$-PO$_4$$^{2-}$ are directly perturbed by isotope-labelling, which reduces the probability of additional, slower rearrangements in the protein required to release the product. Further studies will be necessary to interrogate the contributions from these two models, but the *Ab*ATPPRT results revealed, to the best of our knowledge, a novel facet of the heavy-enzyme isotope effects approach where mass-dependent effects manifest themselves as physical steps regulated by fast dynamics.

## Methods

### Reagents
All commercially available chemicals were used without further purification. BaseMuncher endonuclease was purchased from AbCam. Ampicillin, dithiothreitol (DTT), isopropyl $\beta$-D-1-thiogalactopyranoside (IPTG) and 2-(*N*-morpholino)ethanesulfonic acid-sodium dodecyl sulfate (MES-SDS) were purchased from Formedium. DH5$\alpha$ chemically competent *Escherichia coli*, DpnI were purchased from New England Biolabs (NEB). QIAprep Spin Miniprep was from Qiagen. Ethylenediaminetetraacetic acid (EDTA)-free Complete protease inhibitor cocktail was from Roche. ATP, ADP, BL21(DE3) chemically competent *E. coli*, D$_2$O, glycerol, histidine, imidazole, lysozyme, PRPP, KCl, MgSO$_4$, MgCl$_2$, MnCl$_2$, CaCl$_2$, KH$_2$PO$_4$, Na$_2$HPO$_4$, NaCl, NH$_4$Cl, [$^{15}$N]NH$_4$Cl, D-glucose, [$^{13}$C$_6$]D-glucose, [$^{13}$C$_6$,$^2$H$_7$]D-glucose, [$^2$H$_7$]D-glucose, BME vitamins, and tricine were from Merck. Agarose, dNTPSs, kanamycin, 4-(2-hydroxyethyl)piperazine-1-ethanesulfonic acid (HEPES), PageRuler Plus Prestained protein ladder, PageRuler$^{TM}$ Plus Prestained protein ladder, and SYPRO orange protein gel stain were from ThermoFisher Scientific. DNA oligonucleotide primers were synthesised by Integrated DNA Technologies (IDT). *Mt*PPase and tobacco etch virus protease (TEVP) were produced as previously described[29]. *Ab*HisZ was produced as published[30]. *Ab*HisG$_S$ was produced from LB culture as published[30]. PRATP was produced as previously described[62].

### Site-directed mutagenesis of *Ab*HisG$_S$
Generation of R70A-*Ab*HisG$_S$ by site-directed mutagenesis was carried out with overlapping primers according to the method of Liu and Naismith[63]. Forward and reverse primer sequences were 5′-GATTCTGGC CGCCTCTGATGTTCCGACGTACGTTGAAAACG-3′ and 5′-GGCG GCCAGAATCAGGATACGCACCTGTTTGTGGG-3′, respectively. The WT-*Ab*HisG$_S$ expression vector[30] was used as DNA template. Correct insertion of the mutation was confirmed by DNA sequencing performed by Eurofins Genomics. The unlabelled R70A-*Ab*HisG$_S$ was expressed and purified by the same protocol as the WT-*Ab*HisG$_S$[30], and its concentration was determined spectrophotometrically (NanoDrop) at 280 nm based on the theoretical extinction coefficient ($\varepsilon_{280}$) of 10,430 M$^{-1}$ cm$^{-1}$ as calculated in the ProtParam tool of Expasy.

### *Ab*HisG$_S$ expression in M9 minimum medium
For unlabelled *Ab*HisG$_S$ expression, *E. coli* BL21(DE3) cells transformed with the *Ab*HisG$_S$ expression vector[30] were grown in 1 LB supplemented with 50 µg mL$^{-1}$ kanamycin at 37 °C until an OD$_{600}$ of 0.6–0.8 was reached. Cells were harvested by centrifugation (6774 × *g*, 15 min, 4 °C), rinsed with M9 minimum medium, centrifuged (6774 × *g*, 10 min, 4 °C), before resuspension in 1 L M9 minimum medium[47] in the presence of 50 µg mL$^{-1}$ kanamycin. The culture was equilibrated to 16 °C for 1 h before induction with 0.5 mM IPTG and additional 20-h incubation at 16 °C. Cells were harvested by centrifugation (6774 × *g*, 15 min, 4 °C) and stored at −20 °C. For unlabelled and isotope-labelled *Ab*HisG$_S$ expression by the high-cell-density expression method, the same procedure was followed except an OD$_{600}$ of 3.0–5.0 was reached[47]. The specific isotope labels were introduced to WT- and R70A-*Ab*HisG$_S$ via the high-cell-density expression method by supplementing the M9 minimum media with differently isotope-labelled chemicals as outlined in Supplementary Table 14. Unlabelled and isotope-labelled *Ab*HisG$_S$ variants were purified, and their concentrations determined, as previously described[30]. WT- and R70A-*Ab*HisG$_S$ isotopologues had their intact mass determined by ESI/TOF-MS analysis performed by the

BSRC Mass-Spectrometry and Proteomics Facility at the University of St Andrews.

### DSF
DSF measurements ($\lambda_{ex}$ = 490 nm, $\lambda_{em}$ = 610 nm) for all WT-*Ab*HisG$_S$ isotopologues were performed in 96-well plates on a Stratagene Mx3005p instrument. Reactions (50 µL) contained 100 mM tricine, 100 mM KCl, 15 mM MgCl$_2$, 4 mM DTT pH 8.5, 6 µM enzyme, with 5X Sypro Orange (Invitrogen) added to each well. Thermal denaturation curves were recorded over a temperature range of 25–93 °C with increments of 1 °C min$^{-1}$. DSF measurements ($\lambda_{ex}$ = 520 nm, $\lambda_{em}$ = 558 nm) for all WT-*Ab*ATPPRT isotopologues, in the presence and absence of 250 µM PRATP, and for all R70A-*Ab*HisG$_S$ and R70A-*Ab*ATPPRT isotopologues were performed in 96-well plates on a QuantStudioÔ1 Real-Time PCR instrument. Reactions (20 µL) contained 100 mM tricine, 100 mM KCl, 15 mM MgCl$_2$, 4 mM DTT pH 8.5, 8 µM enzyme, with 5X Sypro Orange (Invitrogen) added to each well. Thermal denaturation curves were recorded over a temperature range of 25–93 °C with increments of 0.05 °C s$^{-1}$. Control curves lacked protein and were subtracted from curves containing protein.

### General assay for enzyme activity under steady-state conditions
Initial rates at 10 °C and above were performed in the forward direction in 100 mM tricine pH 8.5, 15 mM MgCl$_2$, 100 mM KCl, 4 mM DTT, and 10 µM *Mt*PPase. Either PRATP or $N^1$-(5-phospho-$\beta$-D-ribosyl)-ADP (PRADP) formation was monitored by the increase in absorbance at 290 nm ($\varepsilon_{290}$ = 3600 M$^{-1}$ cm$^{-1}$)[64] over 60 s with readings every 1 s in 1-cm path-length quartz cuvettes (Hellma) in a Shimadzu UV-2600 spectrophotometer outfitted with a CPS unit for temperature control. Reactions (500 µL) were incubated for 3 mins at the desired temperature prior to being initiated by the addition of PRPP. Initial rates at 5 °C were obtained by monitoring the increase in absorbance at 290 nm due to either PRATP or PRADP formation in an Applied Photophysics SX-20 stopped-flow spectrophotometer outfitted with a 5-µL mixing cell (0.5-cm path length and 0.9 ms dead time) and a circulating water bath for temperature control. One syringe contained all proteins (*Ab*HisG$_S$, *Mt*PPase, and *Ab*HisZ where applicable) and either ATP or ADP, while the other contained PRPP. Both syringes contained 100 mM tricine pH 8.5, 100 mM KCl, 15 mM MgCl$_2$, and 4 mM DTT. Reactions were triggered by rapidly mixing 55 µL from each syringe and monitored for 60 s. Control reactions lacked PRPP. Furthermore, controls were conducted to ensure rates were independent of *Mt*PPase concentration. This was ascertained empirically by increasing the *Mt*PPase concentration in the assay and confirming the rates did not change.

### General assay for enzyme activity under pre-steady-state conditions
Rapid kinetics under multiple- and single-turnover conditions at 5 and 25 °C were carried out by monitoring the increase in absorbance at 290 nm due to PRATP formation in an Applied Photophysics SX-20 stopped-flow spectrophotometer outfitted with a 5-µL mixing cell (0.5-cm path length and 0.9 ms dead time) and a circulating water bath for temperature control. In every experiment, one syringe contained all proteins (*Ab*HisG$_S$, *Mt*PPase, and *Ab*HisZ where applicable) and either ATP or PRPP, while the other contained either PRPP or ATP. Both syringes contained 100 mM tricine pH 8.5, 100 mM KCl, 15 mM MgCl$_2$, and 4 mM DTT. Reactions were triggered by rapidly mixing 55 µL from each syringe. Control reactions lacked PRPP.

### Apparent equilibrium dissociation constant ($K_D$) for *Ab*HisZ
The $K_D$ for *Ab*HisZ was determined at various temperatures and in the presence and absence of 12% glycerol (v/v) by measuring initial rates of WT-*Ab*HisG$_S$ isotopologues (0.04 µM, except for [$^{13}$C,$^{15}$N]*Ab*HisG$_S$ at any temperature, and for unlabelled *Ab*HisG$_S$ at 35 °C, whose concentration was 0.02 µM) in the presence of 1.4 mM ATP, 1.0 mM PRPP, and varying concentrations of *Ab*HisZ (0–0.5 µM, except at 35 °C in 12% glycerol, where the range was 0–0.2 µM). For R70A-*Ab*HisG$_S$ isotopologues (0.24 µM at 25 °C, 0.30 at 5 °C, and 0.30 µM at 25 °C in 12% glycerol) in the presence of

1.4 mM ATP, 1.0 mM PRPP (or 1.6 mM ATP and 1.4 mM PRPP in 12% glycerol), and varying concentrations of *Ab*HisZ (0–1 μM at 25 °C, 0–4 μM at 5 °C).

### Histidine dose-dependence for *Ab*ATPPRT
The histidine dose-dependence on *Ab*ATPPRT isotopologues was determined as previously published[30].

### Substrate saturation curves for *Ab*HisG$_S$ at 5, 25, and 40 °C
Initial rates for *Ab*HisG$_S$ isotopologues were measured at saturating concentrations of one substrate (either 3.2 mM PRPP or 6.4 mM ATP) and varying concentrations of the other, either ATP (0–6.4 mM) or PRPP (0–3.2 mM). At 5 °C, enzyme concentrations were 3 μM for unlabelled *Ab*HisG$_S$, [$^{15}$N]*Ab*HisG$_S$, and [$^2$H,$^{13}$C,$^{15}$N]*Ab*HisG$_S$, and 1.5 μM for [$^{13}$C,$^{15}$N]*Ab*HisG$_S$. At 25 °C, enzyme concentrations were 1 μM for unlabelled *Ab*HisG$_S$ and [$^{15}$N]*Ab*HisG$_S$, and 0.75 μM for [$^{13}$C,$^{15}$N]*Ab*HisG$_S$ and [$^2$H,$^{13}$C,$^{15}$N]*Ab*HisG$_S$. At 40 °C, enzyme concentration was 0.25 μM for unlabelled *Ab*HisG$_S$ and [$^2$H,$^{13}$C,$^{15}$N]*Ab*HisG$_S$. In addition, at 40 °C, *Ab*HisG$_S$ (0.1 μM) substrate saturation curves were also determined in the presence of 15 mM MnCl$_2$ instead of MgCl$_2$.

### Substrate saturation curves for *Ab*ATPPRT at 5 and 25 °C
Initial rates for *Ab*ATPPRT isotopologues were measured at saturating concentrations of one substrate (1.6 mM either PRPP, ATP or ADP) and varying concentrations of the other, either the nucleotide (0–1.6 mM) or PRPP (0–1.6 mM). With ATP as substrate at 5 °C, enzyme concentrations were 0.039 μM for unlabelled *Ab*ATPPRT, [$^{15}$N]*Ab*ATPPRT, [$^2$H,$^{15}$N]*Ab*ATPPRT and [$^2$H,$^{13}$C,$^{15}$N]*Ab*ATPPRT, and either 0.014 μM (at high substrate concentration) or 0.028 μM (at low substrate concentration) for [$^{13}$C,$^{15}$N]*Ab*ATPPRT. At 25 °C, enzyme concentrations were 0.080 μM for unlabelled *Ab*ATPPRT, [$^{15}$N]*Ab*ATPPRT, [$^2$H,$^{15}$N]*Ab*ATPPRT and [$^2$H,$^{13}$C,$^{15}$N]*Ab*ATPPRT, and 0.040 for [$^{13}$C,$^{15}$N]*Ab*ATPPRT. With ADP as substrate at both temperatures, 0.040 μM enzyme was used.

### R70A-*Ab*HisG$_S$ activity at 25 °C
Initial rates for 10 μM R70A-*Ab*HisG$_S$ were measured for 5 min in the presence of 3.2 mM PRPP, 6.4 mM either ATP or ADP, and 25 μM *Mt*PPase. Controls lacked enzyme.

### Substrate saturation curves for R70A-*Ab*ATPPRT at 5 and 25 °C
Initial rates for *Ab*ATPPRT isotopologues were measured at saturating concentrations of one substrate (either 3.2 mM PRPP or 6.4 mM ATP or ADP) and varying concentrations of the other, either the nucleotide (0–6.4 mM) or PRPP (0–3.2 mM). At 5 °C, enzyme concentration was 0.500 μM. At 25 °C, enzyme concentration was 0.240 μM.

### Solvent viscosity effects for *Ab*ATPPRT
At 5 °C, initial rates were determined in the presence 10 μM *Mt*PPase, of 1.4 mM and 1.6 mM PRPP, 1.6 mM either ATP or ADP, and 0–12% glycerol (v/v). WT-*Ab*ATPPRT concentration was 0.080 μM with ATP as substrate, and 0.040 μM with ADP as substrate. At 25 °C, initial rates were determined in the presence of 3.0 mM and 3.2 mM PRPP, 6.4 mM ATP, and 0–12% glycerol (v/v). R70A-*Ab*ATPPRT concentration was 0.230 μM. At 35 °C, initial rates were determined in the presence of 1.6 mM and 2.0 mM PRPP, 1.6 mM ATP, and 0–12% glycerol (v/v). WT-*Ab*ATPPRT concentration was 0.020 μM. Initial rates were also measured in the presence of 15 μM *Mt*PPase at the highest PRPP concentration and 12% glycerol to confirm the rates were not dependent on *Mt*PPase concentration. At 5 °C with ADP as substrate, and at 35 °C with ATP as substrate, where a significant solvent viscosity effect was observed, the rates were also measured in the presence of 5% PEG-8000.

### Temperature-rate profile for *Ab*ATPPRT
The temperature stability of *Ab*ATPPRT was assessed by incubating 0.025 μM *Ab*ATPPRT, 10 μM *Mt*PPase, 1.6 mM ATP at either 30 °C, 35 °C, or

40 °C for 10 min before incubating at 30 °C for 3 min and measuring initial rates upon addition of PRPP to a final concentration of 1.6 mM. The temperature-rate profile was determined by measuring initial rates of *Ab*ATPPRT at temperatures ranging from 5 °C (278 K) to 35 °C (308 K) in 5-°C increments at saturating concentrations of one substrate (1.6 mM either PRPP or ATP) and varying concentrations of the other substrate near saturation at each temperature (0.4–1.6 mM). *Ab*ATPPRT concentrations were 0.040 μM for 5 °C and 25 °C, 0.050 μM for 10 °C and 15 °C, and 0.025 μM for the remaining temperatures.

### Temperature-rate profile for *Ab*HisG$_S$
The temperature-rate profile was determined by measuring initial rates of *Ab*HisG$_S$ at temperatures ranging from 5 °C (278 K) to 45 °C (318 K) in 5-°C increments at saturating concentrations of one substrate (either 3.2 mM PRPP or 6.4 mM ATP) and varying concentrations of the other substrate near saturation at each temperature (0.8–3.2 mM PRPP; 1.6–6.4 mM ATP). *Ab*HisG$_S$ concentrations were 3 μM for 5 °C, 1 μM for 10 °C to 25 °C, 0.5 μM for 30 °C, 0.25 μM for the remaining temperatures.

### *Ab*HisG$_S$ and *Ab*ATPPRT rate-dependence on KCl concentration
*Ab*HisG$_S$ initial rates (at 25 °C and 40 °C) were measured in the presence of 50–150 mM KCl, 3.2 mM PRPP, 6.4 mM ATP, and either 1 μM (25 °C) or 0.250 μM (40 °C) *Ab*HisG$_S$. *Ab*ATPPRT initial rates (at 25 °C and 35 °C) were measured in the presence of 50–150 mM KCl, 1.6 mM PRPP, 1.6 mM ATP, and either 0.040 μM (25 °C) or 0.025 μM (35 °C) *Ab*ATPPRT. Between the lowest and highest temperatures, the maximum pH variation of the buffer was 0.2 units, and the assay pH of 8.5 lies on a pH-independent region of the *Ab*ATPPRT pH-rate profile[46].

### Temperature dependence of the *Ab*ATPPRT $^{HE}k_{cat}$
Initial rates of *Ab*ATPPRT isotopologues were measured from 5 to 25 °C in 5 °C increments in the presence of 1.6 mM of each substrate and 0.040 μM enzyme except for [$^{13}$C,$^{15}$N]*Ab*ATPPRT, whose concentration was 0.030 μM.

### Multiple-turnover pre-steady-state kinetics
Rapid kinetics of PRATP formation by *Ab*HisG$_S$ isotopologues at 5 and 25 °C were carried out as previously described[46], except that at 25 °C, 2000 data points were collected in 2.5 s.

### Single-turnover pre-steady-state kinetics
Rapid kinetics of PRATP formation by *Ab*ATPPRT isotopologues under single-turnover conditions at 5 °C was performed as previously reported[46], except that enzyme concentration used here was 80 μM. Rapid kinetics of PRATP formation by R70A-*Ab*ATPPRT under single-turnover conditions at 25 °C was performed with enzyme concentrations of 80 μM and 100 μM for unlabelled *Ab*ATPPRT and 100 μM [$^{13}$C,$^{15}$N]R70A-*Ab*ATPPRT with 4700 data points collected in 0.47 s, with 6 μM PRPP and 6.4 mM ATP.

### Multiple sequence alignment of HisG$_S$
Sequences for the multiple sequence alignment were acquired by first conducting a BLASTp search of the UniProtKB/Swiss-Prot database using the *Ab*HisG$_S$ sequence as the query. Sequences showing 100% sequence identity to *Ab*HisG$_S$ were excluded. For *Ab*HisG$_S$ this returned a total of 256 ATP phosphoribosyltransferase sequences; those containing over 250 residues were excluded, as it is likely that those corresponded to HisG$_L$ sequences, resulting in a total of 205 different HisG$_S$ sequences. The MSA was conducted using Clustal Omega[65]. The degree of conservation was calculated using the AL2CO algorithm[50] as implemented in ChimeraX[66] then mapped onto the structure of the *Ab*HisG$_S$ protein (PDB: 8OY0)[46].

### Kinetics and thermal denaturation data analysis
Kinetics and thermal denaturation data were analysed by the nonlinear regression function of SigmaPlot 14.0 (SPSS Inc.). Thermal denaturation data were fitted to Eq. (1). Substrate saturation curves at a fixed

concentration of the co-substrate were fitted to Eq. (2). Initial rate data at varying concentrations of HisZ were fitted to Eq. (3). The concentration of ATPPRT at any concentration of $Ab$HisG$_S$ and $Ab$HisZ was calculated according to Eq. (4). Histidine dose-response data were fitted to Eq. (5). Single-turnover data with WT-$Ab$ATPPRT isotopologues were fitted to Eq. (6). Plots of $k_{cat}$ ratios against relative viscosity were fitted to Eq. (7). Temperature-rate profile for $Ab$HisG$_S$ was fitted to Eq. (8). Single-turnover data with R70A-$Ab$ATPPRT isotopologues were fitted to Eq. (9). In Eqs. (1)–(9), $F_U$ is fraction unfolded, $T$ is the temperature in °C, $T_m$ is the melting temperature, $c$ is the slope of the transition region, and $LL$ and $UL$ are folded and unfolded baselines, respectively; $k_{cat}$ is the steady-state turnover number, $v$ is the initial rate, $E_T$ is total enzyme concentration, $K_M$ is the apparent Michaelis constant, $S$ is the concentration of the varying substrate, $V_{max}$ is the maximal velocity, $G$ is the concentration of $Ab$HisG$_S$, $Z$ is the concentration of $Ab$HisZ, $K_D^{app}$ is the apparent equilibrium dissociation constant, $ATPPRT$ is the concentration of $Ab$ATPPRT holoenzyme, IC$_{50}$ is the half-maximal inhibitory concentration of inhibitor, $n$ is the Hill coefficient, $I$ is the concentration of inhibitor, $v_i$ and $v_0$ are initial rates in the presence and absence of inhibitor, $P(t)$ is product concentration as a function of time $t$, $k_2$ and $k_3$ are rate constants governing sequential steps in a single turnover, ES is the enzyme-substrate complex concentration, $k_{cat}^0$ and $k_{cat}^\eta$ represent the $k_{cat}$ in the absence and presence of glycerol, respectively, $\eta_{rel}$ is the relative viscosity of the solution, $m$ is the slope, $k_{STO}$ is the apparent single-turnover rate constant, $A$ is the signal amplitude, $k_B$, $h$, and $R$ are the Boltzmann, Planck, and gas constants, respectively, $T$ is the temperature, $T_0$ is the reference temperature (298 K here), $\Delta H_{T_0}^\ddagger$ and $\Delta S_{T_0}^\ddagger$ are the activation enthalpy and entropy, respectively, at $T_0$, and $\Delta C_P^\ddagger$ is the activation heat capacity.

$$F_U = LL + \frac{UL - LL}{1 + e^{(T_m - T)/c}} \tag{1}$$

$$\frac{v}{E_T} = \frac{k_{cat}S}{K_M + S} \tag{2}$$

$$v = V_{max}\frac{G + Z + K_D^{app} - \sqrt{(G + Z + K_D^{app})^2 - 4GZ}}{2G} \tag{3}$$

$$ATPPRT = \frac{G + Z + K_D^{app} - \sqrt{(G + Z + K_D^{app})^2 - 4GZ}}{2} \tag{4}$$

$$\frac{v_i}{v_0} = \frac{1}{1 + \left(\frac{I}{IC_{50}}\right)^n} \tag{5}$$

$$P(t) = \frac{ES}{k_2 + k_3}\left[k_2\left(1 - e^{-k_3 t}\right) - k_3\left(1 - e^{-k_2 t}\right)\right] \tag{6}$$

$$\frac{k_{cat}^0}{k_{cat}^\eta} = m\left(\eta_{rel} - 1\right) + 1 \tag{7}$$

$$\ln\frac{k_{cat}}{T} = \ln\frac{k_B}{h} - \frac{\Delta H_{T_0}^\ddagger + C_P^\ddagger(T - T_0)}{RT} + \frac{\Delta S_{T_0}^\ddagger + C_P^\ddagger\ln(T/T_0)}{R} \tag{8}$$

$$P(t) = A\left(1 - e^{-k_{STO}t}\right) \tag{9}$$

## Reporting summary

Further information on research design is available in the Nature Portfolio Reporting Summary linked to this article.

## Data availability

All protein mass spectrometry data, all kinetics and DSF data, and the full sequence alignment data were deposited to FigShare under DOI 10.6084/m9.figshare.24631194 [https://doi.org/10.6084/m9.figshare.24631194].

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

## Acknowledgements

This work was supported by the Biotechnology and Biological Sciences Research Council (BBSRC) [Grant BB/M010996/1] via an EASTBIO Doctoral Training Partnership studentship to B.J.R. The authors thank the BSRC Mass-Spectrometry and Proteomics Facility at the University of St Andrews for the ESI/TOF-MS analyses.

## Author contributions

B.J.R. carried out all experimental work, analysed data, and co-wrote the manuscript. J.B.O.M. analysed data. R.G.d.S. conceived and supervised the research, analysed data, and co-wrote the manuscript.

## Competing interests

The authors declare no competing interests.
