## [Peer Review File · Communications Chemistry]

Reviewers' comments:

Reviewer #1 (Remarks to the Author):

This is an interesting yet highly complex manuscript, and I find it hard to assess whether the combination of carefully designed and executed experiments make coherent insights into the energetics and role of dynamics in this enzyme.

Their enzyme of choice is an interesting system - a heteroenzyme with separate catalytic and allosteric chains. They have previously determined that the activation of the enzyme in the full complex is due to a change in the rate determining step. Here they seek to probe the role of dynamics in the chemical step by using a heavy labelled catalytic subunit and observing the effects on the kinetic profile - both carrying out steady-state measurements (at variable temperature) and fast kinetics. There are complexities - to see burst in the complex they need to lower the temperature. Likewise, they demonstrate that the protein is unstable.

There are a series of interesting and somewhat perplexing experimental results and while there is some attempt to make broader conclusions about the energetics of the reaction and propose that rate determining conformational rearrangements govern the chemistry. But a variety of possible interesting explanations are provided for their results.

One of the biggest limitations is that they are attempting to make generalised conclusions about this form of ATPPRT b - it is difficult to assess whether the details of catalysis and allostery that are uncovered here pertain for more than the enzyme from *Acinetobacter baumannii*. Indeed the abstract does claim these general conclusions without referencing this experimental constraint.

It would be fascinating to see equivalent experiments of another variant of this enzyme - in the absence of that I think the generality of the conclusions need to be stated with less hype.

Reviewer #2 (Remarks to the Author):

What are the major claims of the paper?

The authors reported the unique mass-dependent effects on AbATPPRT because the heavy-enzyme kinetic isotope effects are evident exclusively upon allosteric activation of the enzyme, attributed to PRATP release, a rate-limiting protein motion at lower temperature only.

Are they novel and will they be of interest to others in the community and the wider field?

If the conclusions are not original, it would be helpful if you could provide relevant references. Yes, there are original conclusions.

Is the work convincing, and if not, what further evidence would be required to strengthen the conclusions? Yes, the data strongly supports the claims in the manuscript.

I am not sure whether it is a word-to-pdf conversion issue, somehow throughout the manuscript, all the labels in the figures are in very thick font, making the superscript of H, C and N not easy to read. Also the colours chosen for the 4 datasets overlaid are very similar on my screen, except the purple curves.

Fig1. Consider completing the 1a into a mechanism, such as ATP's N1 nucleophilic attacks the C1 with a curly arrow and label the substrates and products with the abbreviations used in the manuscript. Also the Mg²⁺ should not be under the arrow of the reverse reaction, making chemists think it only participates in the reverse direction.

If KCl affects the catalytic rate, I'd assume that in the manuscript this is mentioned first and all the following measurements are done with KCl, but this aspect was not mentioned until Page15. I also looked into the method and cannot find out whether this is the case.

Since Mn²⁺ is more efficient for the charge balance, why it is not used in all the assays?

Some ATP or ADP are delivered as Li⁺ salt. What is the counter ion for their ATP and ADP? If not potassium or sodium, does it affect the catalytic rates? Could the author comment on this?

Page 9, can the authors clarify what the type of isomerisation they were referring to in "an isomerisation of the AbATPPRT:ATP:PRPP complex", or are they referring to the conformational changes in the arrangement of the active site? As they later mentioned "this isomerisation may involve a vibration motion" If so, what evidence do they have?

Some minor points:

Table 1. Please check the significant number for the value $(8 \pm 1) \times 10^3$.

Fig3. Consider labelling the temperatures to 3b and 3c, making the comparison easier.

Fig4. Consider labelling the residue numbers to arginines in 4a and 4b, also label the temperature to 4c and 4d, making the comparison easier.

Page 5, abbreviation AbATPPRT is mentioned twice.

Page 17 (PRADP) should be PRATP

Page 22 (kSTO of 42 \pm 1) is missing the unit

Page 23 "Mass modulation of such low-frequency motions are not readily intuitive, since the typically small increase in protein mass (e.g. 5% – 11%) would lead to an imperceptible decrease in vibration frequency due to negligibly low restoring force constants" Can the authors add a couple of references to explain and clarify this claim?

Page 33 first line 5- C. typo.

Reviewer #3 (Remarks to the Author):

In the article by Read, Mitchell and da Silva a heavy isotope effect that is discovered for the allosteric enzyme ATP phosphoribosyltransferase is described. The data and analysis are of high quality and the topic is of general interest to the fields of biochemistry and enzymology. The main finding is that a clear HE effect is observed for AbATPPRT at 5C while absent at 25 C. These data are described in figures 1 and 2. The finding is related to a previous observation (ref #44) that a step after chemistry is rate limiting for AbATPPRT. The remaining sections of the manuscript are aimed at finding possible explanations to the observed HE. In short no direct explanation for the HE is found. However, the experiments related to figures 4 & 5 e are of high quality and very interesting and collectively with the initial findings warrants publication in Communications Chemistry. The authors found that the HE is abolished when ADP rather than ATP is used as substrate and that the HE (potentially and see major points below) is abolished with ATP for the Arg70Ala variant. Rather than trying to link explain the data (discussion section) with weak linkages through coupling of fast (ps-ns) and slow (us-ms) motion I suggest that the authors more directly discuss their findings. In summary I recommend publication after major revision.

Major points

The paper hinges on the conclusion from ref#44 that a step after chemistry is rate limiting for AbATPPRT. I suggest that the evidence in support of this conclusion is summarized in the introduction.

In the main text and the M & M section it is not clear how the allosterically activated species is produced namely mixing of HisG with HisZ, it is mentioned on page 8 and should be clearly stated early in the introduction.

I found the results linked to Figure 3 redundant in the context of the search for an explanation to the finding of mass dependent k_{cat} at 5 C and I suggest that the authors consider removing this part of the manuscript.

In the findings related to figure 4 it is demonstrated that ADP results in a k_{cat} is increased relative to that of ATP. When using ADP is P_i rather than PP_i released from the enzyme? If so, is it possible that release of P_i follows different kinetics compared to PP_i ?

The authors conclude that there is no HE effect to k_{cat} for the Arg70 variant at 25C but in Fig 5e it seems there exist a limited HE effect? I ask the authors to comment on this

I would encourage the authors to scale down on the number of items in the supplement.

The conclusion is a bit "hand waiving" and I suggest a stronger focus on the findings in the manuscript.

Minor points:

In the legend to figure 1 replace "never" with "not". Also in figure 1e the authors can indicate the PRPP and ATP concentrations used.

In the legend to figure 4 the temperature can be indicated in the title of the figure.

Reviewer #1

There are a series of interesting and somewhat perplexing experimental results and while there is some attempt to make broader conclusions about the energetics of the reaction and propose that rate determining conformational rearrangements govern the chemistry. But a variety of possible interesting explanations are provided for their results. One of the biggest limitations is that they are attempting to make generalised conclusions about this this form of ATPPRT b - it is difficult to assess whether the details of catalysis and allostery that are uncovered here pertain for more than the enzyme from *Acintobacter baumannii*.

Authors' response: We agree with the reviewer that our results should not be generalised to other forms of ATPPRT in lieu of experimental evidence for those other forms. We do not believe we have generalised our results to other ATPPRTs, either short- or long-forms. In fact, we were careful throughout the manuscript (except for the abstract, please see the next comment/reply) to be specific to the orthologue we are working with. As an example, on page 5, the paragraph starts with:

“For *A. baumannii* ATPPRT (*AbATPPRT*), unique among other reported ATPPRTs due to its reaction proceeding via a rapid equilibrium random mechanism,³¹ steady-state and pre-steady-state kinetics studies point to chemistry as the rate-limiting step for nonactivated HisG_S (henceforth referred to as *AbHisG_S*). Allosteric activation by HisZ (*AbHisZ*)...”

And from this point onwards, we use the prefix “*Ab*” to specify that we are working on the orthologue from *A. baumannii*.

The final paragraph of the introduction (page 6), where we set out our initial hypothesis and aims, continues to abide by the constraint that we are working with the “*Ab*” orthologue, as it reads:

“We hypothesize fast protein dynamics are involved in the significant allosteric enhancement of the chemical step in *Ab*ATPPRT, which would be susceptible to protein-mass modulation. We employed various isotope-labelling patterns of *Ab*HisG_S accompanied by differential scanning fluorimetry (DSF), alternative-substrate kinetics, site-directed mutagenesis, steady-state and pre-steady-state enzyme kinetics, and temperature-rate profiles to probe the effect of increased protein mass on *Ab*ATPPRT catalysis and allostery.”

This deliberate specification of the orthologue of ATPPRT continues without exception throughout the Results and Discussion sections. Whenever our analysis touched on other ATPPRTs or other enzymes, we made clear the boundaries of our results. For example, at the end of the paragraph at the top of page 25, referring to the conservation of residues across HisG_S, we cautioned:

“...While there are hotspots of amino acid sequence conservation outside the active site in HisG_S (**Figure 5a**), their involvement in enabling PRATP release is yet unknown.”

Indeed the abstract does claim these general conclusions without referencing this experimental constraint.

Authors' response: To make clear from the start that the results refer to *A. baumannii* ATPPRT, we modified the first part of the abstract to read:

“Heavy-isotope substitution into enzymes slows down bond vibrations and may alter transition-state barrier crossing probability if this is coupled to fast protein motions. ATP phosphoribosyltransferase from *Acinetobacter baumannii* is a multi-protein complex where the regulatory protein HisZ allosterically enhances catalysis by the catalytic protein HisG_S. This is accompanied by a shift in rate-limiting step from chemistry to product release. Here we report...”

It would be fascinating to see equivalent experiments of another variant of this enzyme - in the absence of that I think the generality of the conclusions need to be stated with less hype.

Authors' response: We agree with the reviewer vis-à-vis interrogating this approach to other ATPPRT orthologues, to test whether or not this is a general feature of this enzyme family. In fact, we stated something along those lines in the manuscript (page 24), but we were careful to state that similar studies would have to be carried out with these other enzymes before conclusions could be reached:

“...One might hypothesise, by analogy to the alkaline phosphatase proposal, the unaltered mass of Mg^{2+} prevents coupling from fast protein motions to chemistry in *Ab*ATPPRT. Expanding these studies to other ATPPRTs and other metalloenzymes may provide a test of this hypothesis.”

In summary, we could not find evidence in the manuscript that we unduly extrapolated our results on *Ab*ATPPRT, or even implied as much, to other ATPPRT orthologues, much less other enzymes.

Reviewer #2

I am not sure whether it is a word-to-pdf conversion issue, somehow throughout the manuscript, all the labels in the figures are in very thick font, making the superscript of H, C and N not easy to read. Also the colours chosen for the 4 datasets overlaid are very similar on my screen, except the purple curves.

Authors' response: It is possible it is a word-to-pdf conversion issue, as the reviewer alluded to. Separate high-resolution figures will be submitted along with the revised manuscript, for final publication, should the manuscript be accepted. The colours on the figures were chosen in consideration of recommendations of colours suitable for most readers, avoiding specially green and red.

Fig1. Consider completing the 1a into a mechanism, such as ATP's N1 nucleophilic attacks the C1 with a curly arrow and label the substrates and products with the abbreviations used in the manuscript. Also the Mg^{2+} should not be under the arrow of the reverse reaction, making chemists think it only

participates in the reverse direction.

Authors' response: We have changed Fig 1a to depict the chemical mechanism of the reaction. The Mg^{2+} is part of the chelated complex with reactants/products, and the enzyme abbreviation is omitted.

If KCl affects the catalytic rate, I'd assume that in the manuscript this is mentioned first and all the following measurements are done with KCl, but this aspect was not mentioned until Page15. I also looked into the method and cannot find out whether this is the case.

Authors' response: All activity measurements contain 100 mM KCl in the reaction, except for those testing the effect of varying KCl concentration, which contain different concentrations of the salt. The presence of KCl in all functional assays is state in the Methods under the subheadings "DSF", "General assay for enzyme activity under steady-state conditions", and "General assay for enzyme activity under pre-steady-state conditions", which cover basically all assays. To make sure the dependence of ATPPRT activity on KCl is state upfront, we have added the following sentence to the introduction (page 4):

"ATPPRT activity is also dependent on KCl.²⁹⁻³³"

Since Mn^{2+} is more efficient for the charge balance, why it is not used in all the assays?

Authors' response: Because Mg^{2+} is the physiological divalent metal in this reaction; thus we only employ Mn^{2+} as a useful probe of the mechanism/rate-limiting step. The increase in *AbHisGs* activity with Mn^{2+} should be considered in light of the massive excess of Mg^{2+} over Mn^{2+} typically found in bacterial cells (~1,000-fold excess). ATPPRT is not a metalloenzyme, but an enzyme whose substrates (PRPP and ATP) and products (PRATP and PPi) are present in their Mg^{2+} -chelated form in a cell.

Some ATP or ADP are delivered as Li⁺ salt. What is the counter ion for their ATP and ADP? If not potassium or sodium, does it affect the catalytic rates? Could the author comment on this?

Authors' response: All ATP and ADP employed in the work comes as the disodium salt.

Page 9, can the authors clarify what the type of isomerisation they were referring to in “an isomerisation of the *Ab*ATPPRT:ATP:PRPP complex”, or are they referring to the conformational changes in the arrangement of the active site? As they later mentioned “this isomerisation may involve a vibration motion” If so, what evidence do they have?

Authors' response: There are two distinct isomerisation processes here. In the first one, on page 9, as quoted by the reviewer, isomerisation is used as a general term, as is common in the context of enzyme complexes, for a kinetically detectable change in the *Ab*ATPPRT:ATP:PRPP ternary complex before on-enzyme PRATP (which is what produces the absorbance signal we are detecting in the stopped-flow spectrophotometer) is formed. This is inferred from the lag time in the single-turnover experiments depicted in Figure 2c (which we had observed before, published in reference 46). We do not know the nature of this isomerisation. It may be, for instance, a change in protonation state of a group, and it may as well be a conformational change of the ternary complex, with the corollary that it is protein-mass independent. To give an example of what the term isomerisation might include, we modified the sentence in question on page 9 to:

“...probably an isomerisation (e.g. a conformational change) of the *Ab*ATPPRT:ATP:PRPP complex followed by on-enzyme formation of PRATP.”

The other isomerisation the reviewer is referring to is an isomerisation of the *Ab*ATPPRT:PRATP complex (enzyme-product complex), represented as EQ = EQ* in Figure 2f, that precedes diffusional departure of PRATP from the enzyme. This isomerisation is invoked, as explained in the manuscript, from the following evidence: k_{cat} is limited by a step after chemistry both 25 °C and 5 °C (from burst

kinetics in reference 46, and single-turnover kinetics here), and from our previous work on *Ab*ATPPRT and *P. arcticus* ATPPRT, PP_i has negligible affinity for *Ab*ATPPRT:PRATP complex and dissociates fast. However, unlike the case at 25 °C, where PRATP diffusion from the enzyme is rate-determining as evidenced by massive solvent viscosity effects, at 5 °C there is no solvent viscosity effect, indicating that a step after chemistry and likely after PP_i departure, therefore involving the *Ab*ATPPRT:PRATP complex, precedes PRATP diffusion from the enzyme and limits k_{cat} . The assertion that this isomerisation is likely a vibrational motion involving the *Ab*ATPPRT:PRATP complex comes from the observation that k_{cat} (limited by this isomerisation step) is reduced as the isotopic mass of the enzyme increases.

Table 1. Please check the significant number for the value $(8 \pm 1) \times 10^3$.

Authors' response: We rechecked that value. Dividing k_{cat} ($2.27 \pm 0.06 \text{ s}^{-1}$) by $K_{\text{M}}^{\text{ATP}}$ ($0.00027 \pm 0.00004 \text{ M}$) and propagating the error yields $8,407 \pm 1,265 \text{ M}^{-1} \text{ s}^{-1}$; thus the uncertainty lies before the first decimal place, and the final number is represented in scientific notation as $(8 \pm 1) \times 10^3 \text{ M}^{-1} \text{ s}^{-1}$. There is no point in adding a decimal figure since this would be undercut by the uncertainty.

Fig3. Consider labelling the temperatures to 3b and 3c, making the comparison easier.

Authors' response: We have added the respective experimental temperature directly to Figures 3b and 3c, and also to Figure 3g.

Fig4. Consider labelling the residue numbers to arginines in 4a and 4b, also label the temperature to 4c and 4d, making the comparison easier.

Authors' response: We have added the respective experimental temperature directly to Figures 4c and 4d, and also labelled the R73 residues in the *P. arcticus* ATPPRT structures in Figure 4a, and the R70 and R73 (from *A. baumannii* and *P. arcticus* ATPPRTs, respectively) in Figure 4b.

Page 5, abbreviation AbATPPRT is mentioned twice.

Authors' response: We have removed the first "*AbATPPRT*".

Page 17 (PRADP) should be PRATP

Authors' response: Actually, it should be PRADP; what was incorrect was the description preceding the abbreviation, which should end with -ADP instead of ATP. We have now fixed this in the sentence:

"...the presumed absence of this interaction when N^1 -(5-phospho- β -D-ribose)-ADP (PRADP) is the product..."

Page 22 (kSTO of 42 \pm 1) is missing the unit

Authors' response: We have added the unit (s^{-1}).

Page 23 "Mass modulation of such low-frequency motions are not readily intuitive, since the typically small increase in protein mass (e.g. 5% – 11%) would lead to an imperceptible decrease in vibration frequency due to negligibly low restoring force constants" Can the authors add a couple of references to explain and clarify this claim?

Authors' response: We had cited references 53 and 55 at the end of the paragraph, and those are the references for the sentence in question. To make this clear, we have added the citations at the end of the sentence in question as well:

"Mass modulation of such low-frequency motions are not readily intuitive, since the typically small increase in protein mass (e.g. 5% – 11%) would lead to an imperceptible decrease in vibration frequency due to negligibly low restoring force constants.^{53,55}"

Page 33 first line 5- C. typo.

Authors' response: We changed it now to “5 °C”.

Reviewer #3

The paper hinges on the conclusion from ref#44 that a step after chemistry is rate limiting for AbATPPRT. I suggest that the evidence in support of this conclusion is summarized in the introduction.

Authors' response: We had outlined (bottom of page 5) some of the key evidence from reference 46 (previously reference 44), which is now fully published, for our conclusion that a step after chemistry, specifically product release, limits *AbATPPRT* k_{cat} . We have modified the sentence slightly to make clear how the k_{STO} contributes to that conclusion. The modified sentence now reads:

“On the other hand, with *AbATPPRT*, a burst of product formation was inferred, although it was too fast at 25 °C to observe directly even with rapid kinetics, suggesting a step after chemistry is rate-limiting. This was corroborated by high solvent viscosity effects, which showed PRATP diffusion from the enzyme to be rate-determining for k_{cat} .⁴⁶ At 5 °C, the burst phase could finally be observed with *AbATPPRT*; moreover, the single-turnover rate constant (k_{STO}), which was much higher than k_{cat} , showed the chemical step is allosterically activated more than 1,300-fold in *AbATPPRT* as compared with *AbHisGs*.⁴⁶”

In the main text and the M & M section it is not clear how the allosterically activated species is produced namely mixing of HisG with HisZ, it is mentioned on page 8 and should be clearly stated early in the introduction.

Authors' response: The holoenzyme is generated *in vitro* by mixing the two proteins at defined amounts. To make this point clear from the beginning, we added the following to the last paragraph of the introduction:

“Taking advantage of the fact that *AbHisG_s* and *AbHisZ* are purified independently, and the *AbATPPRT* holoenzyme generated *in vitro* by mixing the two proteins at defined concentrations,^{30...}”

I found the results linked to Figure 3 redundant in the context of the search for an explanation to the finding of mass dependent *k_{cat}* at 5 °C and I suggest that the authors consider removing this part of the manuscript.

Authors' response: The results from Figure 3b-d are not part of our search for the molecular origins of the observed isotope effect, but to test the system further with an additional isotope combination, and to uncover more detail how the isotope effects change with temperature, as opposed to just 5 °C and 25 °C. Figure 3e-g present data that are part of our investigation of the origins of the isotope effect. We believe therefore it is important to leave Figure 3 as it is.

In the findings related to figure 4 it is demonstrated that ADP results in a *k_{cat}* is increased relative to that of ATP. When using ADP is *P_i* rather than *PP_i* released from the enzyme? If so, is it possible that release of *P_i* follows different kinetics compared to *PP_i*?

Authors' response: Regardless of whether ATP or ADP is used, *PP_i* is always the co-product. This is because the *PP_i* comes from the PRPP, not from the nucleotide. To avoid confusion, Figure 1a now has the *PP_i* and the *PP_i* moiety of PRPP in pink to convey that the *PP_i* comes from PRPP.

The authors conclude that there is no HE effect to *k_{cat}* for the Arg70 variant at 25C but in Fig 5e it seems there exist a limited HE effect? I ask the authors to comment on this.

Authors' response: This might seem to be the case at first inspection of the saturation curves in Figure 5e. However, the small difference in k_{cat} values ($2.17 \pm 0.06 \text{ s}^{-1}$ vs $2.01 \pm 0.04 \text{ s}^{-1}$; Supplementary Table 13) between the isotopologues is not statistically significant by a Student's t-test ($p > 0.15$), as reported in Supplementary Table 13.

I would encourage the authors to scale down on the number of items in the supplement.

Authors' response: We believe all the results/analyses presented as supplementary information must be presented.

The conclusion is a bit “hand waiving” and I suggest a stronger focus on the findings in the manuscript.

Authors' response: In the Results section, we provided the most direct interpretation and conclusion for every experiment. In the Discussion section, we contextualised these results in relation to other key heavy-enzyme isotope effects published in the literature (first paragraph of the Discussion), and we summarise the key findings from our work, discuss their significance, and compare them with the literature (second and third paragraphs of the Discussion). At the end of the third paragraph, and in the final two paragraphs we present and discuss hypotheses that might explain our results. That is not “hand waving”. We clearly framed the ideas discussed in the text as hypotheses, and categorically stated further experimental evidence is required to test them.

Minor points:

In the legend to figure 1 replace “never” with “not”.

Authors' response: Done.

Also in figure 1e the authors can indicate the PRPP and ATP concentrations used.

Authors' response: The concentrations were 1.4 mM ATP and 1.0 mM PRPP. This is now stated in the legend to Figure 1e.

In the legend to figure 4 the temperature can be indicated in the title of the figure.

Authors' response: Actually, more than one temperature are employed in the results shown in Figure 4. We now show the temperature on the graphs in Figure 4c and d.

REVIEWERS' COMMENTS:

Reviewer #1 (Remarks to the Author):

The authors have made some improvements to the manuscript. I appreciate the clarification to A. baumannii made now in the abstract. I also note that they have more carefully referred to their previous work which provides evidence on the rate determining step - this is crucial to the interpretation of results and the conclusions presented in the manuscript.

Reviewer #2 (Remarks to the Author):

I am happy with the reply that the authors provided in the rebuttal. Thus I don't have any more comments.

Reviewer #3 (Remarks to the Author):

First, I want to state that the findings in the manuscript have merit and could be presented in a manner that makes it generally interesting and accessible. The results on isotope effects definitely have the potential to move the field of enzymology forward. However, I find that the authors only have made limited efforts to improve the manuscript and the revised version is more or less identical to the initial submission. The data is presented in a way that makes the manuscript difficult to read with too much data and figures that does not add to the main message. Therefore, I regret to convey that I do not find the revised version improved to the degree that warrants publication.

Reviewer #1

The authors have made some improvements to the manuscript. I appreciate the clarification to A. baumannii made now in the abstract. I also note that they have more carefully referred to their previous work which provides evidence on the rate determining step - this is crucial to the interpretation of results and the conclusions presented in the manuscript.

Authors' response: We thank for reviewer for their assessment.

Reviewer #2

I am happy with the reply that the authors provided in the rebuttal. Thus I don't have any more comments.

Authors' response: We thank for reviewer for their assessment.

Reviewer #3

First, I want to state that the findings in the manuscript have merit and could be presented in a manner that makes it generally interesting and accessible. The results on isotope effects definitely have the potential to move the field of enzymology forward. However, I find that the authors only have made limited efforts to improve the manuscript and the revised version is more or less identical to the initial submission. The data is presented in a way that makes the manuscript difficult to read with too much data and figures that does not add to the main message. Therefore, I regret to convey that I do not find the revised version improved to the degree that warrants publication.

Authors' response: We have nothing else to add.